# Evaluation of WRF and CHIMERE models for the simulation of PM$_{2.5}$ in large East African urban conurbations.

Andrea Mazzeo[1,2], Michael Burrow[1], Andrew Quinn[1], Eloise A. Marais[3], Ajit Singh[2], David Ng'ang'a[4], Michael J. Gatari[4], and Francis D. Pope[2]

*1. School of Civil Engineering, University of Birmingham, Birmingham UK*
*2. School of Geography Earth and Environmental Sciences – GEES, University of Birmingham, Birmingham UK*
*3. Department of Geography, University College London, London, UK.*
*4. Institute of Nuclear Science and Technology, University of Nairobi, Nairobi, Kenya*
.

*Correspondence to* Andrea Mazzeo (a.mazzeo@bham.ac.uk )

**Abstract:** *Urban conurbations of East Africa are affected by harmful levels of air pollution. The paucity of local air quality networks and the absence of capacity to forecast air quality make difficult to quantify the real level of air pollution in this area. The chemistry-transport model CHIMERE has been used along with the meteorological model WRF to run simulations at high spatial resolution (2×2 km) of hourly concentrations of Particulate Matter PM$_{2.5}$ for three East African urban conurbations: Addis Ababa in Ethiopia, Nairobi in Kenya, and Kampala in Uganda. Two existing emission inventories were combined to test the performance of CHIMERE as an air quality model for a target monthly period of 2017 and the results compared against observed data from urban, roadside, and rural sites. The results show that the model is able to reproduce hourly and daily temporal variability of aerosol concentrations close to observations in urban, roadside and in rural environments. CHIMERE's performance as a tool for managing air quality was also assessed. The analysis demonstrated that despite the absence of high-resolution data and up-to-date biogenic and anthropogenic emissions, the model was able to reproduce 66 – 99% of the daily PM$_{2.5}$ exceedances above the WHO 24-hour mean PM$_{2.5}$ guideline (25 µg m$^{-3}$) in the three cities. An analysis of the 24-hour average levels of PM$_{2.5}$ was also carried out for 17 constituencies in the vicinity of Nairobi. This showed that 47% of the constituencies in the area exhibited poor air quality index for PM$_{2.5}$ in the unhealthy category for human health exposing between 10,000 to 30,000 people/km$^2$ to harmful levels of air contamination.*

**Keywords:** Air quality, East Africa, Particulate Matter, Anthropogenic emissions, numerical modelling, Air Quality Index

## 1 Introduction

The world's population has grown rapidly by 1 billion people in the last 12 years, reaching 7.9 billion in 2021. The World Population Prospects (WPP) made by the United Nations (U.N.) suggest a continuing annual increase of 1.8 %, meaning the global population will reach 8.5 billion by 2030, 9.7 by 2050, and 11.2 by 2100 (UN-WPP, 2019). The African continent is predicted to have the fastest growing population rate in the world, and it is projected to double between 2010 and 2050, surpassing two billion (UN-WPP, 2019). In addition to this a 60 % increase in population has been predicted by 2050, specifically in urban areas (UN-WPP, 2019).

Population in Sub-Saharan East African (SSEA) countries have increased drastically from 1991 to 2019. In that
period of time and according to data from the World Bank database (WB, 2022), the Kenyan population grew
from 24 to 52 million, the Ugandan population from 17 to 44 million and the Ethiopian population from 50 to 112
million. These increases in population were accompanied by a similar rate of increase in road transport, industrial
activities and in the use of solid fuels (e.g., woods, charcoal, and agricultural residues) for cooking purposes in
urban areas (Bockarie et al., 2020;Marais et al., 2019).
As a result of these population increases, air quality of the urban areas of these countries, historically influenced
by the large presence of seasonal burning biomass emissions (Haywood et al., 2008;Lacaux, 1995;Liousse et al.,
2010;Thompson A. M., 2001), is progressively degrading (Marais and Wiedinmyer, 2016). This, in combination
with the expanding urban population, has greatly increased the exposure of citizens to harmful Particulate Matter
(PM) pollution with an aerodynamic diameter smaller than 10 and 2.5 µm ($PM_{10}$ and $PM_{2.5}$, respectively) (Gatari
et al., 2019;Kinney et al., 2011;Li et al., 2017;UN-Habitat, 2017).
Several diseases have been attributed to PM exposure in SSEA, including cardiovascular and cardiopulmonary
diseases, cancers, and respiratory deep infections (Dalal et al., 2011;Mbewu, 2006;Parkin et al., 2008). In 2012,
the World Health Organization (WHO) estimated 176,000 deaths in SSEA were directly connected to air pollution
(WHO, 2012). Modelling studies have also found that exposure to outdoor air pollution has led to 626,000
disability-adjusted life per year (DALYs) in SSEA alone (Amegah and Agyei-Mensah, 2017), highlighting that
these numbers could be much higher considering the limited amount of air quality data emanating from the region
that are available for research purposes.
Considering the likely severe impacts of air pollution on human health in SSEA, the research interest in
understanding air pollution trends in East Africa has increased in recent years. Many researchers have analysed
the levels of contamination by short-term measurement campaigns (Amegah and Agyei-Mensah, 2017;deSouza
P., 2017;Egondi et al., 2013;Gaita et al., 2014;Gatari et al., 2019;Kume, 2010;Ngo et al., 2015;Pope et al.,
2018;Schwander et al., 2014;Vliet, 2007;Singh et al., 2021). Other studies observed annual average $PM_{2.5}$
concentrations in the order of 100 µg m$^{-3}$ quantified in a small number of urban areas of SSEA (Brauer et al.,
2012).  These levels are about four times higher than the 24-hour average and ten times higher than the annual
average WHO guidelines for $PM_{2.5}$ (Avis W. and Khaemba W., 2018;WHO, 2016) and underline that air pollution
is a serious problem in this area of the world. A recent study by Singh et al. (2020), using visibility as a proxy for
PM, showed that air quality in Addis Ababa, Kampala and Nairobi has degraded alarmingly over the last 4
decades.
The lack of long-term air quality monitoring networks in many African countries have made it difficult to have
reliable long-term air quality data (Petkova, 2013;Pope et al., 2018;Singh et al., 2020) and still little is known
about the levels of air contamination in large urban conurbations (Burroughs Peña and Rollins, 2017). The paucity
and sometimes complete absence of reliable data on air pollution levels makes it difficult to quantify the magnitude
of the problem. Consequently, it is difficult for local and national authorities to plan possible improvement
measures for the mitigation of anthropogenic emissions. Even if important steps forward have been made to

improve the knowledge relative to anthropogenic emissions and emission inventories for Africa used for numerical simulations and forecasts of air quality (Assamoi and Liousse, 2010;Liousse, 2014;Marais and Wiedinmyer, 2016) the lack of surface observations to validate the emission magnitude and the simulated concentrations make these inventories susceptible of large error.

In this work we test a meteorological and a chemistry-transport model (CTM) to simulate the hourly urban and rural levels of $PM_{2.5}$ in three SSEA urban conurbations during a monthly period of 2017. We present the results of the validation of both models for the capital cities of Kenya, (Nairobi), Ethiopia (Addis Ababa) and Uganda (Kampala) against observation data. For Nairobi, we compare model outputs with observations from rural and roadside sites observations collected during the "A Systems approach to Air Pollution in East Africa" research project (ASAP-East Africa - www.asap-eastafrica.com, hereafter called ASAP) (Pope et al., 2018). For Addis Ababa and Kampala, the model was validated using hourly observations of $PM_{2.5}$ collected by the respective U.S. Embassies.

Moreover, we assess the suitability of the CTM as a decision support tool for policy makers to plan possible mitigation policies oriented to quantify the real level of air pollution in urban areas and quantify the human exposure to $PM_{2.5}$. Specifically, in terms of the accuracy of the model we estimate the daily WHO threshold limit exceedances of $PM_{2.5}$ in the three urban conurbations. Finally, for the particular case of Nairobi, we evaluate the average air quality indices by local constituency for the whole analysed period giving a new insight of the real level of air contamination in Nairobi to the general public and the relative population exposed to harmful level of air contamination.

**2 Material and Methods**

The meteorological and chemistry-transport models used in this work have been configured to simulate hourly weather parameters and concentrations of $PM_{2.5}$ using available input data for the simulations and observations from the real world for the validation. The availability of the observations for the validation of both models comes from different providers, have different frequency in time and, in the case of $PM_{2.5}$ observations, come from different environments (rural, urban, roadside sites). No vertical observations were available for the validation of both models.

**2.1 Meteorological model WRF**

The Weather Research and Forecasting (WRF) model is a numerical model for weather predictions and atmospheric simulations and is used commercially and for research purposes, including by the US National Oceanic and Atmospheric Administration (Powers, 2017;Skamarock, 2008).

WRF was used to drive the meteorology for CHIMERE using three geographical domains at different resolutions (from 18×18 km to 2×2 km) vertically divided into 30 levels, nine of which are below 1500 m. The first external domain has a spatial resolution of 18×18 km (Figure 1), with three nested domains at a resolution of 6×6 km

centred on the three countries of interest (Figure 1, white squares).  Three further nested domains with a resolution
of 2×2 km centred on Addis Ababa, Kampala, and Nairobi (Figure 1, white dashed squares, and Figure 2) are the
focus of the analysis.

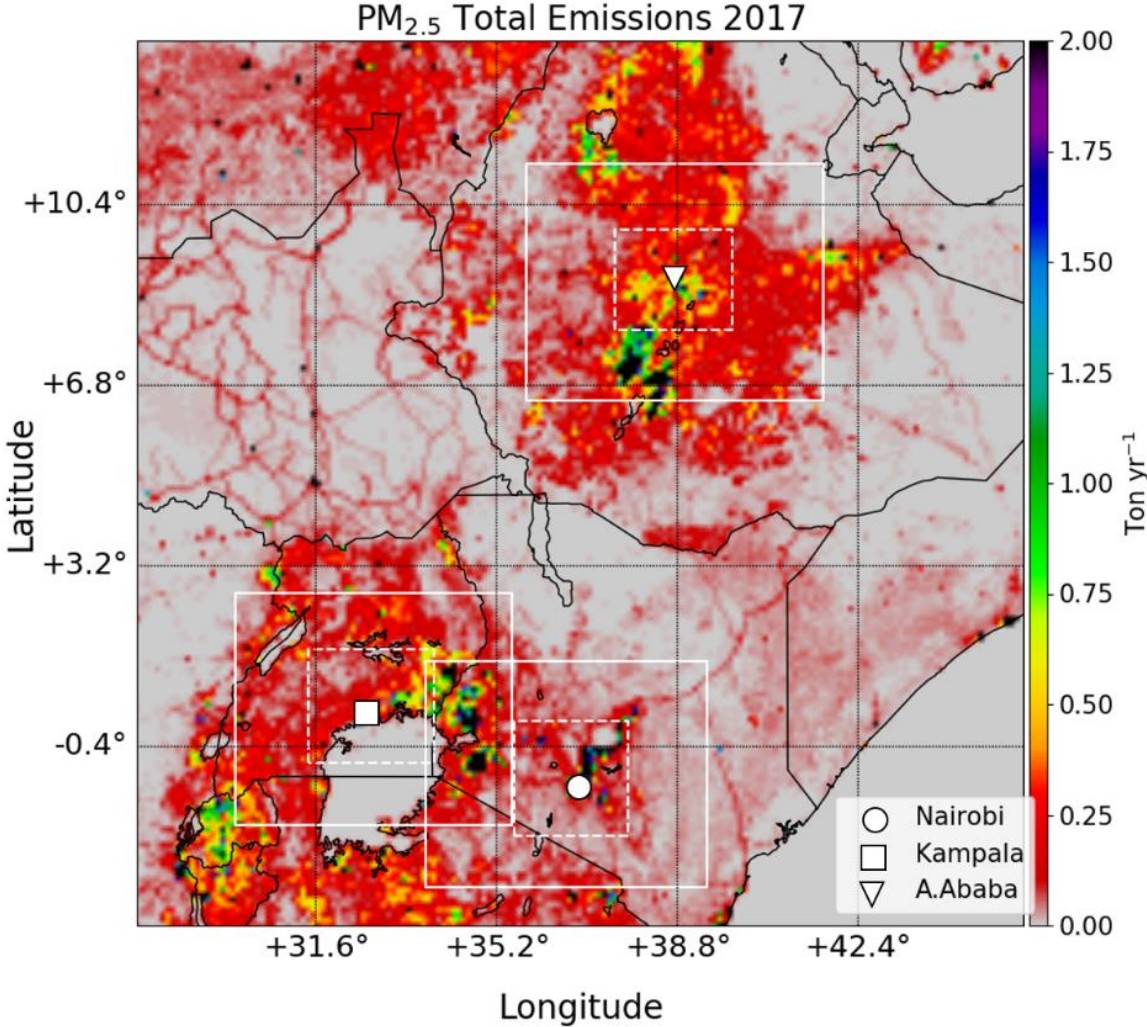


**Figure 1:** *Spatial distribution of the PM₂.₅ emissions from DICE-EDGAR merged emission inventory for East Africa for the*
*WRF domain at 18×18 km of resolution. The continuous white lines show the location of the first nested domain at 6×6 km of*
*resolution used in WRF-CHIMERE. The dashed white squares give the locations of the second nested domains at 2×2 km*
*centred on Addis Ababa (Ethiopia, white triangle), Kampala (Uganda, white square) and Nairobi (Kenya, white circle) used*
*for WRF-CHIMERE.*

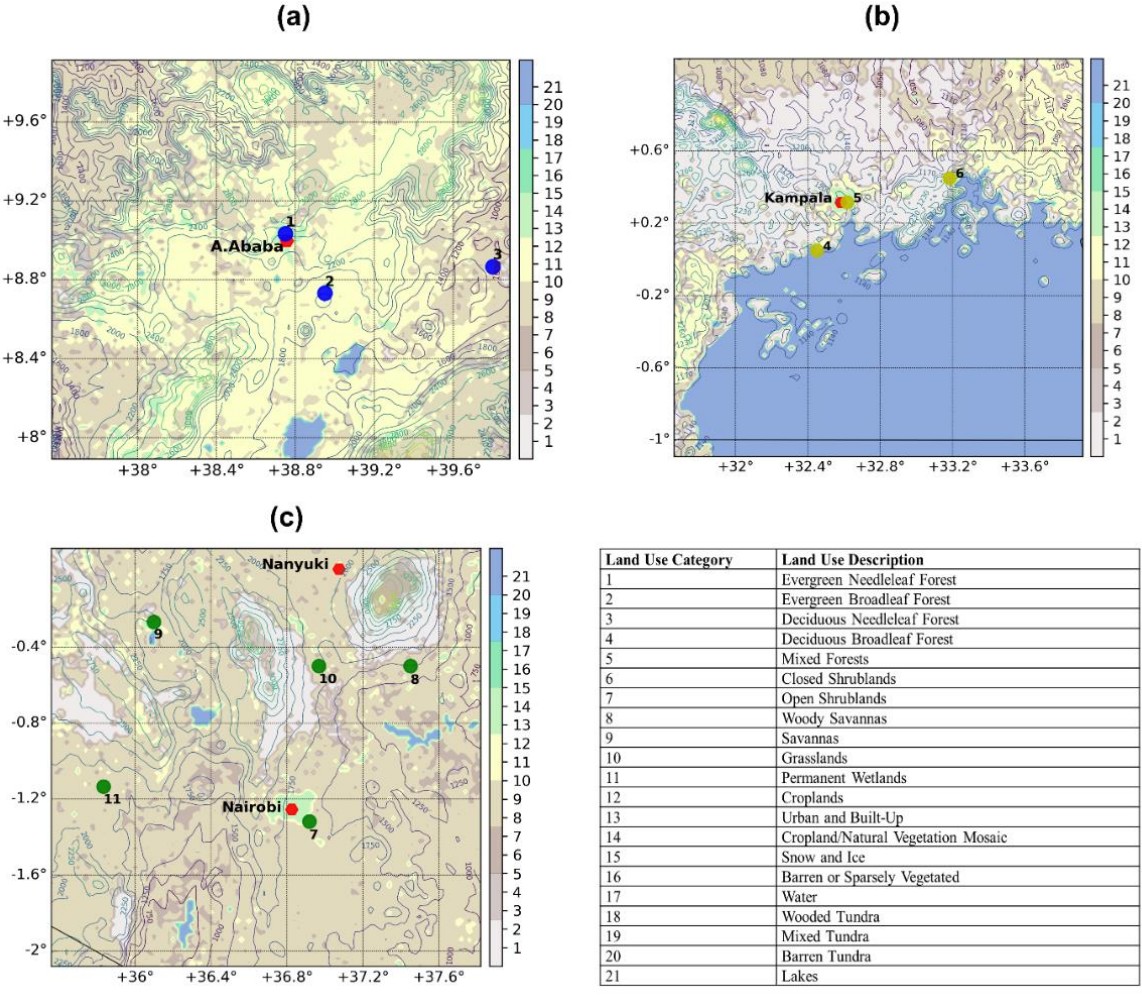


**Figure 2:** *Second-nested domains at a spatial resolution of 2×2 km centred on the cities of Addis Ababa (ETH2K - a), Kampala (UGA2K – b), Nanyuki and Nairobi (KEN2K - c) created using the WRF model outputs. The red dots represent locations of PM$_{2.5}$ measurements. The blue, yellow, and green dots refer to the location of the ground weather stations used for the meteorological validation in Ethiopia, Uganda, and Kenya, respectively. The numbers relate to the stations detailed in Table 2. Contour lines are relative to the height meters from the ground levels from WRF outputs while the colour scale applied to the maps a, b and c represents the 21 classes of classification of the land use adopted in WRF simulations. The description of each land use category is provided the legend.*


The configuration adopted for the WRF simulations has been chosen according to previous works made on East
Africa (Kerandi et al., 2016;Kerandi et al., 2017;Pohl et al., 2011) and is summarized in Table 1. The Yonsei
University Scheme (YSU - (Hong S., 2006)) was chosen to represent the Planetary Boundary Layer while the
Community Atmosphere Model (CAM - (Collins, 2004)) was used for the long and short-wave radiation scheme.
Initial and boundary conditions for the external coarse domain at 18×18 km were obtained from the NCEP FNL
(Final) Operational Global Analysis data (Wu, 2002). Boundary condition for the first (6×6 km) and second (2×2
km) nest domains were taken from the respective parent domains using the two-way-nesting approach. The
process enables the lateral conditions for the internal domains to be calculated from the outputs of the respective
parent domains at lower resolution at every time step of the simulation.

The land use option chosen for the simulations was NOAH (Tewari, 2004) while the WRF Single–moment 3–
class Scheme (WSM3) for clouds and ice proposed by Hong S. (2004) was chosen for the reproduction of the
microphysical processes in WRF.

**2.2 The Chemistry Transport model CHIMERE**

CHIMERE, version 2017r4 (Mailler et al., 2017), is a Eulerian numerical model for reproducing three-
dimensional gas-phase chemistry and aerosols processes of formation, dispersion, wet and dry deposition over a
defined domain with flexible spatial resolutions. CHIMERE has been used for a number of comparative research
studies of Ozone and particulate matter $PM_{10}$ from the continental scale, (Bessagnet et al., 2016;Zyryanov et al.,
2012) to the urban scale (van Loon et al., 2007;Vautard et al., 2007;Mazzeo et al., 2018). Furthermore, the model
has been used for event analysis, scenario studies (Markakis et al., 2015;Trewhela et al., 2019), forecasts, and
impact studies of the effects of air pollution on health (Valari and Menut, 2010) and vegetation (Anav et al., 2011).
The authors highlight that the version of CHIMERE adopted is the 2017r4, the most recent available at the time
when the present work was realized.

CHIMERE model has been used to simulate the first nested domains at 6×6 km and the second nested domains at
2×2 km of spatial resolution. The configuration adopted in this work uses initial and boundary conditions from
the global three-dimensional chemistry-transport model (LMDz-INCA, Hauglustaine et al. (2004)), both for
gaseous pollutants and for aerosols for the most external domain at 6×6 km of resolution while for the most
internal domains at 2×2 km of resolution, the boundary conditions are calculated from model outputs of the parent
domains. The complete chemical mechanism used for all the simulations was SAPRC-07-A (Carter, 2010) which
can describe more than 275 reactions of 85 species. SAPRC-07-A is the most recent chemical mechanism
available for CHIMERE version 2017r4.

Horizontal and vertical diffusion is calculated using the approach suggested by Van Leer (1979) and the
thermodynamic equilibrium ISORROPIA model (Nenes, 1998) is used for the particle/gases partitioning of semi-
volatile inorganic gases. The model permits calculation of the thermodynamical equilibrium between sulphates,
nitrates, ammonium, sodium, chloride and water dependent upon temperature and relative humidity data.

Dry and wet deposition is calculated in CHIMERE. The particle dry deposition velocities are calculated as a
function of particle size and density as well as relevant meteorological variables, including deposition processes,
such as, turbulent transfer, Brownian diffusion, impaction, interception, gravitational settling and particle rebound
(Zhang et al., 2001). Wet deposition is described modelled using a first-order decay equation as described in
Loosmore and Cederwall (2004).

Radiative transfer processes are accounted in CHIMERE using the Fast-JX model (Wild, 2000;Bian, 2002). Fast-
JX is applied also in other models (Voulgarakis, 2009;Real and Sartelet, 2011;Telford et al., 2013). The photolysis
rates calculated by Fast-JX model are validated both inside the limits of the boundary layer (Barnard, 2004) and
in the free troposphere (Voulgarakis, 2009).

Secondary organic aerosols (SOAs), including biogenic and anthropogenic precursors, are modelled in CHIMERE
as described by (Pun, 2006). SOAs formation is represented as a single-step oxidation of the precursors,
differentiating hydrophilic by hydrophobic SOAs in the partitioning formulation. Finally, biogenic emissions were
taken in account within CHIMERE using MEGAN model outputs as described by (Guenther, 2006).

**Table 1:** *Main configuration parameters adopted for the modelling system WRF-CHIMERE for all simulations.*

| WRFv3.9.1 Configuration | | |
|---|---|---|
| **Initial and Boundary conditions** | GFS FNL- Reanalysis | *Wu (2002)* |
| **PBL Parametrization** | YSU | *Hong S. (2006)* |
| **SW/LW Radiation Scheme** | CAM | *Collins (2004)* |
| **Land Use** | NOAH | *Tewari (2004)* |
| **Micro Physics Scheme** | WSM3 | *Hong S. (2006)* |
| **Vertical Levels** | 30 | |
| CHIMERE2017r4 Configuration | | |
| **Initial and boundary conditions** | LMDz-INCA | *Hauglustaine et al. (2004)* |
| **Anthropogenic Emissions** | EDGARv3.4.1 + DICE-Africa | *Crippa M. (2018);Marais and Wiedinmyer (2016)* |
| **Biogenic Emissions** | MEGAN | *Guenther (2006)* |
| **Gas/Aerosol Partitions** | ISORROPIA | *Nenes (1998)* |
| **Secondary Organic Aerosols** | 1 | *Pun (2006)* |
| **Radiative Transfer** | Fast-JX | *Wild (2000);Bian (2002)* |
| **Chemistry Mechanism** | SAPRC-07-A | *Carter (2010)* |
| **Horiz. / Vert. Transport scheme** | VanLeer | *Van Leer (1979)* |
| **Vertical Levels** | 30 | |


**2.3 Emission Inventories**

To correctly describe the impact of anthropogenic emissions on urban air quality of Nairobi, Kampala and Addis
Ababa, industrial and on-grid power generation emissions from the Emissions Database for Global Atmospheric
Research inventory (hereafter EDGAR, version 3.4.1) (Crippa M., 2018) were combined with non-industrial,
prominent combustion sources from the Diffusive and Inefficient Emission inventory for Africa (hereafter DICE)
(Marais and Wiedinmyer, 2016).

EDGAR is a global inventory developed for year 2012 and DICE is a regional inventory for 2013. DICE includes
important sources in Africa (e.g., motorcycles, kerosene use, open waste burning, and *ad hoc* oil refining, among
others) that are absent or misrepresented in global inventories. Both inventories represent the most up-to-date
anthropogenic emissions available for East Africa at the time of the air quality model was used for this work. Both
inventories have spatial resolution of $0.1 \times 0.1°$ and provide annual total of anthropogenic emissions for relevant
gases and aerosols.

On one hand, EDGAR provides emissions data for CO, NO, $NO_2$, $SO_2$, $NH_3$, NMVOCs, BC, OC, $PM_{10}$ and $PM_{2.5}$
as annual totals divided by the sector according to the IPCC-1996 classification. All human activities with
exception of large-scale biomass burning are included in EDGAR (Crippa M., 2018). On the other hand, DICE
provides emissions from particular diffuse and inefficient combustion emission sources (e.g., road transport,
residential biofuel use, energy production and charcoal production and use) for gaseous pollutants (CO, NO, NO$_2$,
SO$_2$, NH$_3$, NMVOCs) and aerosols (BC, OC). Seasonal biomass burning that is considered a large pollution source
in Africa is included in DICE as comparable emissions of black carbon (BC) and higher emissions of nonmethane
volatile organic compounds (NMVOCs). Emissions from DICE were used to provide annual total emissions for
particular emission sources considered to be misrepresented or missing in a global inventory such as EDGAR.
The preparation of the final emission inventory was carried out in two steps. First, DICE and EDGAR inventories
were merged, by pollutant and by sector, following the approach suggested by Marais and Wiedinmyer (2016).
PM$_{2.5}$ emissions are included in DICE as individual components of organic carbon (OC) and black carbon (BC),
but they need to be expressed as lumped PM$_{2.5}$ in CHIMERE. Therefore PM$_{2.5}$ was calculated as the sum of
Organic Carbon (OC - originally present in DICE) multiplied with a conversion factor ($c$ = 1.4) following Pai et
al. (2020) to represent Organic Aerosols emissions  and summed with Black Carbon (BC – originally present in
DICE) as follows:

$$PM_{2.5} = (OC \times c) + BC \qquad\qquad \text{Eq. (1)}$$

Secondly, the emisurf2016 pre-processor of CHIMERE was used to scale the emissions from the original
resolution of 0.1×0.1° (~10 km) to the final resolution of each domain simulated (6×6 and 2×2 km) using
population density data provided from the Socioeconomic Data and Application Centre (SEDAC)
(http://sedac.ciesin.columbia.edu/) as proxy for the spatial distribution. SEDAC provides population density maps
at high resolution (1×1 km) for the years 2010, 2015 and 2020. The SEDAC population density data calculated
for most internal domains at 2×2 km (Figure 2) suggest for 2010 a total population of 7 million for Nairobi, 4.8
million for Kampala and 4.5 million for Addis Ababa. These totals grow respectively to 8.1, 5.9 and 5.0 million
for 2015 and to 9.4, 7.3 and 5.7 million for 2020. The original SEDAC data were used for a linear extrapolation
of the population density data to the target year 2017 and were used by emisurf2016 for the spatial allocation of
the emissions. Additionally, emisurf2016 permitted to temporally distribute the original total annual emissions
rates according to seasonal, weekly, and daily variation profiles. The resulting merged inventory (hereafter, DICE-
EDGAR) totals by pollutant and sectors for the most external domain at 18×18 km of resolution are shown in
Figure 3.
Biogenic emissions and mineral dust considered in this work have been calculated in-line by CHIMERE. The
former are calculated by MEGAN model outputs as described by Guenther (2006) while the latter are calculated
using the USGS land use database provided by CHIMERE. The soil is represented by relative percentages of sand,
silt, and clay for each model cell. The USGS database, called STATSGO-FAO accounts of 19 different soil types
recorded in the global database with native resolution of 0.0083×0.0083°. To have homogeneous datasets, the
STATSGO-FAO data are re-gridded into the CHIMERE simulation grids. For mineral dust emission calculations,
the land use is typically used to provide a desert mask specifying what surface is potentially erodible.

The emissions used in this work might not reflect the true values due to missing emission sources and the mismatch
of the simulated time period and the date of the emission inventories. The lack of up-to-date national emission
inventories collected at a sufficient resolution, in addition to the lack of research sources providing projections of
emissions for 2017, meant that it was not possible to generate more detailed information about the anthropogenic
sources of emissions for East Africa.

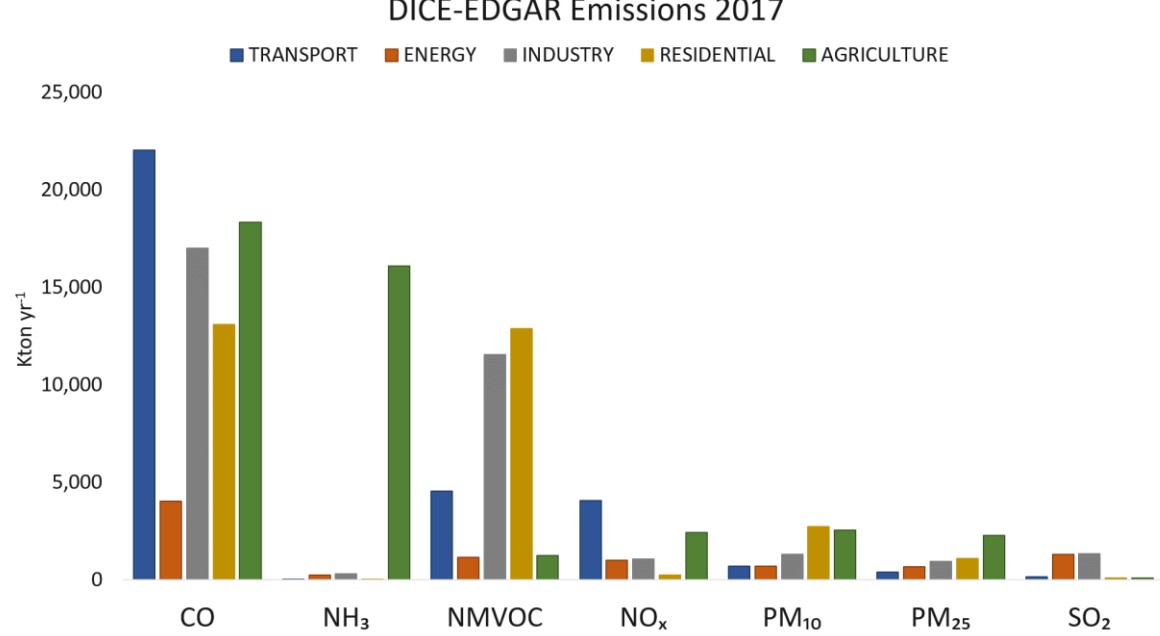


**Figure 3:** *Annual Totals for the merged emission inventory DICE-EDGAR for year 2017 calculated on the spatial domain at*
*18×18 km shown in Figure 1.*

It is noted that the time stamp of the anthropogenic emissions and the validation period are different. The emissions
are relative to year 2013 while the observation used for the validation for 2017. In the absence of additional data
and in the lack of national or local mitigation policies in the three countries we assume that the differences in time
stamp do not make large difference to the emission estimates. More detailed analysis of the emission sources and
the implementation of possible mitigation policies at national and local levels could in future change this situation.

Finally, we recall that one of the main objectives of the present work is to evaluate the performance of WRF and
CHIMERE models in reproduce meteorology and air pollution levels in urban conurbations using the most-up-to-
date available data and giving in this way a new insight on the state of the art of the numerical modelling for air
quality in this area of the world highlighting possible improvements for future works.

**2.4 Weather and Chemistry Observations**

WRF and CHIMERE models have been validated for a limited monthly period between the 14[th] of February and
14[th] of March 2017. The choice of this period is because of the availability of continuous measurements for the
validation of both models. While for the case of WRF observations with frequency variable from 3 to 6 hours are
available from the UK Met Office MIDAS database (MetOffice, 2012) for different locations, rarer are $PM_{2.5}$
observations that last over one month with a measurement frequency of one hour, and from different environments
(e.g., rural, urban, or roadside sites).

The period chosen for the simulations of meteorology has to be representative of the average weather conditions
of the analysed area and avoid unusual weather conditions (e.g., extreme events) that could impact the physical
and chemical processes described in the CTM and affect the final concentrations of secondary pollutants
simulated. The February to March time period in East Africa does not have extreme temperatures (mean
temperatures approximately 10 - 25°C according to the country) and little rainfall that could affect the observations
of weather conditions and $PM_{2.5}$ concentrations (USAID, 2022). These conditions and the absence of alternative
data covering a large time frame for the validation of CHIMERE have constrained the period of simulation to the
present period.

**Table 2:** UK Met Office *ground weather stations used for the validation of the 2×2km domains. Station n. corresponds to the*
*position of each station in Figure 2a, b and c and PM2.5 observation points for the urban domains of Addis Ababa, Kampala*
*and Nairobi used for the validation of CHIMERE model.*

| Station n. | Domain | Name | Latitude | Longitude | Elevation |
|---|---|---|---|---|---|
| 1 | ETH2K | Addis – Bole | 0.03° N | 38.75° E | 1900 m |
| 2 | | Harar Meda | 8.73° N | 38.95° E | 2355 m |
| 3 | | Metehara | 8.87° N | 39.90° E | 930 m |
| | | U.S. Embassy ($PM_{2.5}$ – urban background) | 9.05° N | 38.76° E | 1900 m |
| 4 | UGA2K | Entebbe (Airport) | 0.05° N | 32.45° E | 1155 m |
| 5 | | Kampala | 0.32° N | 32.62° E | 1144 m |
| 6 | | Jinja | 0.45° N | 33.18° E | 1175 m |
| | | U.S. Embassy ($PM_{2.5}$ – urban background) | 0.30° N | 32.59° E | 1150 m |
| 7 | KEN2K | Nairobi (Airport) | 1.32° S | 36.92° E | 1624 m |
| 8 | | Embu | 0.50° S | 37.45° E | 1493 m |
| 9 | | Nakuru | 0.27° S | 36.10° E | 1901 m |
| 10 | | Nyeri | 0.50° S | 36.97° E | 1759 m |
| 11 | | Narok | 1.13° S | 35.83° E | 2104 m |
| | | Tom Mboya Street ($PM_{2.5}$ – roadside) | 1.28° S | 36.82° E | 1795 m |
| | | Nanyuki ($PM_{2.5}$ – rural background) | 0.01° N | 37.07° E | 1947 m |


Observations of temperature, wind speed and directions used for the validation of WRF were taken from the UK
Met Office MIDAS database. Data from 11 weather stations, three for the domain of Ethiopia (hereafter ETH2K,
Figure 2a) and Uganda (hereafter UGA2K, Figure 2b) and five for the domain of Kenya (hereafter KEN2K,
Figure, 2c) were used to validate the simulations at a resolution of 2×2 km (Table 2).

The ground stations are at different altitudes above sea level to a maximum of 2355 m (e.g., the Harar Meda
station in Ethiopia, n2 in Figure 2a). The validation was performed by comparing model outputs with observations
for the variables, namely surface temperature, wind speed and direction and relative humidity. The latter, not
originally available in the MIDAS dataset, was calculated using the coefficients proposed by Alduchov O. (1996)
based on hourly surface and dew point temperatures observed values and then compared with modelled data
obtained by WRF.

Hourly concentrations of PM$_{2.5}$ were used for the validation of CHIMERE for the three internal domains at 2×2
km (Figure 2). For the city of Nairobi, data from roadside background site located at Tom Mboya Street was used
(1.28° S, 36.82° E), while data from the rural background were provided by a site located in Nanyuki, Kenya
(0.01° N, 37.07° E). Both the field sites data were obtained from the field sampling campaign performed by Pope
et al., (2018). For the urban background locations of Addis Ababa and Kampala, hourly concentration of PM$_{2.5}$
were obtained from the air quality monitoring stations of the two U.S. Embassies in Ethiopia (9.05° N, 38.76° E)
and Uganda (0.30°N, 32.59° E) using optical counters. Data from Uganda and Ethiopia were used to compare the
configuration applied to CHIMERE for Kenya with the two other countries (Table 2).

**2.5 Statistical Parameters**

In this work we use different statistical operators to evaluate the performance of WRF and CHIMERE models in
reproducing the main surface weather parameters and hourly and daily concentrations of PM$_{2.5}$ in different urban
and rural environments. The statistical analysis both for WRF and for CHIMERE has been done calculating the
statistics for each station individually and the averaging all station together. The calculation has been done on the
original hourly values from observations and model outputs and consider hourly values from the model only if
the corresponding hourly observation is present. The statistical parameters of Pearson's Coefficient (R, Eq. 2),
index of agreement (IOA, Eq. 3), mean fractional bias (MFB, Eq. 4) and mean fractional error (MFE, Eq. 5) have
been used for the calculations:

$$R = \frac{n(\sum_{i=1}^{n} M_i O_i) - (\sum_{i=1}^{n} M_i)(\sum_{i=1}^{n} O_i)}{\sqrt{[n\sum_{i=1}^{n} M_i^2 - (\sum_{i=1}^{n} M_i)^2][n\sum_{i=1}^{n} O_i^2 - (\sum_{i=1}^{n} O_i)^2]}}$$    Eq. (2)

$$IOA = 1 - \left[\frac{\sum_{i=1}^{n}(O_i - M_i)^2}{\sum_{i=1}^{n}(|M_i - \bar{O}| + |O_i - \bar{O}|)^2}\right]$$    Eq. (3)

$$MFB = \frac{1}{n}\sum_{i=1}^{n}(M_i - O_i)/((O_i + M_i)/2)$$    Eq. (4)

$$MFE = \frac{1}{n}\sum_{i=1}^{n}|M_i - O_i|/((O_i + M_i)/2)$$    Eq. (5)

MFB and MFE in particular, are metrics specifically used for the evaluation of numerical system for atmospheric
chemistry and meteorology. They normalise the bias and the error for each model-observed pair by the average
of the model and observation before taking the final average. The advantage of these metrics is that the maximum
bias and errors are bounded, and that impact of outlier data points are minimised. Moreover, the metrics are
symmetric giving equal weight, to concentrations simulated higher than observations and to those that are
simulated lower than observations.

MFB and MFE have been expressed in terms of model performance "goals" and model performance "criteria"
values according to the methodology proposed by Boylan and Russell (2006). The performance "goal" for the

modelling system is attested for MFE ≤ 50% and MFB ≤ ± 30%. In this range of the performance of the model in reproducing the correct magnitude of the concentrations can be considered good. A second larger range of values, called "criteria", is attributed for MFE ≤ 75% and MFB ≤ ± 60%. Values inside this are corresponds to an average model performance. Finally, values with MFE > 75% and -60% > MFB > +60% correspond to a poor representation by the model.

**2.6 Model Resolution and Simulations design**

WRF and CHIMERE models run at spatial resolutions of 18×18, 6×6 and 2×2 km for meteorology and at 6×6 and 2×2 km for chemistry for the three domains of East Africa. The statistical analysis shown in the following sections though, describes the validation results for the three internal domains at a resolution of 2×2 km as these are the focus of the present work.

Ground weather stations from the MIDAS database, included in the 2×2 km domains of all countries, were analysed individually, and shown as average of all stations. The time series and wind roses are relative to the closest stations from MIDAS database to each urban city centre of the three capital cities, namely Addis- Bole (n1 in Table 2), Kampala (n5 in Table 2) and Nairobi Airport (n7 in Table 2).

Initially, the performance of CHIMERE was analysed for the domain of Kenya for which hourly concentrations of $PM_{2.5}$ were taken from two different sites (roadside and rural) from the field sampling campaign described by Pope et al., (2018). Secondly, the same configuration adopted for Kenya was used for Ethiopia and Uganda to test both the homogeneity of the emission rates on other urban conditions, and the configuration chosen for CHIMERE in different urban and environmental conditions. At this stage of the validation, a threshold limit of 25 µg m$^{-3}$ for $PM_{2.5}$ per day provided by WHO (WHO, 2005) was used to quantify the number of exceedances observed and modelled by CHIMERE for the three cities.

The validation process was hindered by the highly variable quantity and quality of available meteorological data. The majority of the weather observations are provided on a 3-hourly basis, with varying amounts of missing data. Despite this, the statistical evaluation of WRF has been performed comparing model and observations only when the latter were available. We recall that the objective of this work aims to test the performances of a modelling system for the simulation of air quality at high resolution for East Africa, updating and/or using the available input data available and assessing the possible adoption of these tools for air quality policy making at this extent of the data.

**3 Results and Discussion**

**3.1 Validation of the WRF simulations**

In order to assess the performance of WRF in simulating surface temperature, relative humidity wind speed and
direction, the model simulation outputs were compared with all the available ground weather station data available
for the period of analysis, 14th of February to 14th of March 2017.

*3.1.1 Statistical evaluation of WRF performances*

A statistical analysis, in terms of the mean fractional bias (MFB), mean fractional error (MFE), index of agreement
(IOA) and Pearson's coefficient (R), was carried out to compare modelled and observed values for the domain at
2×2 km resolution averaging the observed and modelled values on all the stations present on each domain (Table
3). We recall that the number and location of the stations is variable between the three domains (3 stations for
ETH2K and UGA2K and 5 stations for KEN2K).

The results of the statistical analysis show that WRF is capable of reproducing the mean levels of surface
temperature better for the domain of Ethiopia (ETH2K) and Uganda (UGA2K) with a mean underestimation over
the three domains of 1.4 and 1.5 °C, respectively, then for Kenya (KEN2K) where it shows an underestimation of
4.1 °C. The higher bias in surface temperature found on the average of all five stations of Kenya is though highly
driven by a particular poor representation of this variable at the observation point of Narok (n11 in Figure 2c)
where the bias between model and observations is 10.9 °C. A reason for this bias can be related by the location of
the station that is the one at highest altitude of all the Kenyan weather stations (2104 m a.g.l.). Narok is located
around 140 km west from Nairobi and the high bias in temperature should not have any effect on the levels of
temperature modelled in the capital of Kenya were the bias for the individual station of Nairobi (n7 in Figure 2c)
found was 1.3 °C.

Relative humidity is overestimated by WRF in KEN2K of 0.2 % and underestimated in ETH2K of 6.4 % and in
UGA2K of 7.5 % (Table 3). Wind Speed and directions for the three domains show respectively, the presence of
northern winds in UGA2K correctly captured by the model with a difference of around 4° in comparison with the
observations, an average eastern wind component in KEN2K partially reproduced by the model that allocates the
average wind directions on a more south-eastern component of wind with a difference of around 40.2° while in
ETH2K the average wind direction modelled and observed are closer with a difference of 4.2° on a south-eastern
component of prevailing wind. The observed and modelled wind speeds in UGA2K, KEN2K and ETH2K suggest
an overestimation by the model of 0.9, 0.8 and 0.2 m s$^{-1}$, respectively (Table 3).

The mean fractional error calculated in the three domains is inside the limit of the goal range both for surface
temperature and for relative humidity with values between 30 and 35 for the former and 11 and 27 for the latter
variable.  On the other hand, the values of MFE for wind speed and directions are more variable according to the
domain. While MFE values for wind directions were found inside the criteria range for all domains, for wind
speed only KEN2K and ETH2K are in this range, while the wind speed in UGA2K was found outside the
acceptability range of model performance (Table 3).

The same analysis done taking in account the mean fractional bias shows values in the goal range for surface
temperature for the three domains, overestimated by the model for UGA2K (0.17) and underestimated for ETH
2K (-5.38) and KEN2K (-24.25). Same behaviour was found also for the relative humidity that seems
underestimated in the three domains but with MFB values inside the goal criteria. Finally, wind speed and
directions are found in the goal range of MFB only for ETH2K while KEN2K shows values of both variables in
the criteria range and UGA2K shows wind direction in the criteria range but wind speed outside the acceptability
range of model performance (Table 3).

**Table 3:** *Statistical analysis of relative humidity, surface temperature, wind speed and directions averaged on all the available*
*weather stations for the second nested domains UGA2K, KEN2K and ETH2K at 2×2 km of resolution. Mean observed and*
*modelled values, Pearson's Coefficient (R), index of agreement (IOA), mean fractional bias (MFB) and error (MFE) have*
*been calculated.*

| | Relative Humidity (%) | | | Temperature (ºC) | | |
|---|---|---|---|---|---|---|
| | UGA2K | KEN2K | ETH2K | UGA2K | KEN2K | ETH2K |
| **Observations Mean** | 68.2 | 63.1 | 51.3 | 24.5 | 23.2 | 22.7 |
| **Model Mean** | 60.7 | 63.3 | 44.9 | 23.0 | 19.1 | 21.3 |
| **MFB** | -21.52 | -21.36 | -33.02 | 0.17 | -24.25 | -5.38 |
| **MFE** | 30.08 | 32.25 | 35.56 | 12.50 | 27.94 | 11.34 |
| **IOA** | 0.44 | 0.44 | 0.47 | 0.43 | 0.31 | 0.53 |
| **R** | 0.3 | 0.4 | 0.7 | 0.3 | 0.5 | 0.6 |
| | Wind Direction (degrees) | | | Wind Speed (m s⁻¹) | | |
| | UGA2K | KEN2K | ETH2K | UGA2K | KEN2K | ETH2K |
| **Observations Mean** | 6.8 | 91.5 | 104.0 | 2.5 | 2.7 | 3.5 |
| **Model Mean** | 2.8 | 131.7 | 99.8 | 3.4 | 3.5 | 3.7 |
| **MFB** | 32.02 | -30.57 | -9.94 | 91.25 | 36.83 | 18.89 |
| **MFE** | 62.01 | 70.55 | 60.18 | 94.59 | 54.35 | 50.63 |
| **IOA** | 0.39 | 0.40 | 0.46 | 0.26 | 0.41 | 0.31 |
| **R** | 0.3 | 0.2 | 0.2 | 0.1 | 0.5 | 0.4 |


The calculated Pearson's coefficient (R) shows the capability of the model in reproducing the minimum and
maximum peaks of different variable values. The R values were found varying between 0.1 and 0.7 for the three
domains. The reproduction of the maximum and minimum values of relative humidity is better in ETH2K where
R value was found approximately 0.7 while the lowest R values occurred in UGA2K (0.3). A similar trend was
found also in the description of the surface temperature with maximum and minimum better reproduced in ETH2K
(0.6), followed by KEN2K (0.5) and UGA2K (0.3). For wind speed, the highest R coefficient value was for
KEN2K (0.5) and the lowest for UGA2K (0.1) while for wind directions, the highest R value found was for
UGA2K (0.3) with values of approximately 0.2 for the other two domains (Table 3).

Finally, the evaluation of the index of agreement (IOA) shows values for surface temperature between 0.31
(KEN2K) and 0.53 (ETH2K) and values between 0.44 and 0.47 for relative humidity in the three domains. For
wind speed and directions, the IOA varies between 0.39 (UGA2K) and 0.46 (ETH2K) for the former and between
0.26 (UGA2K) and 0.41 (KEN2K) for the latter. The comparison of the Index of Agreement between the three
domains suggests that the model performance is higher in reproducing drier areas corresponding to ETH2K and
KEN2K in comparison with the UGA2K where the influence of the Lake Victoria seems to impact the overall
statistical analysis. More variable is the performance of WRF in reproducing the general conditions of wind speed
and directions between the three domains.

*3.1.2 Hourly variation of Temperature and Relative humidity*

The three Met Office stations providing weather observations closest to the urban areas of the Addis Ababa,
Kampala and Nairobi have been analysed individually in form of hourly time series of surface temperature and
relative humidity and wind roses for wind speed and directions.

The hourly surface temperature and relative humidity are shown in Figure 4 for the three ground weather stations
closest to the centre of the three cities: Addis-Bole (n1 in Figure 2a), Kampala Station (n5 in Figure 2b) and
Nairobi (n7 in Figure 2c).

The temperature range observed at the three stations was between 9 and 27° C for the Addis-Bole Station, 16 and
31° C for Kampala and 16 and 33° C for Nairobi. By inspection of Figure 4, it can be seen that the WRF model
is able to reproduce the main diurnal cycle of variation of temperature and relative humidity for the three ground
weather stations. Surface temperature peaks are slightly underestimated by the model for the three stations with a
small mean bias at the three stations between -0.06 and -0.1° C. The highest agreement between the model and
observation is for Kampala while the model tends to underestimate the diurnal peaks of surface temperature almost
systematically for Addis-Bole and Nairobi stations.

The mean relative humidity observed at the three stations shows different ranges of excursion from the model
predictions depending on the characteristics of the environment. The station of Addis-Bole shows the higher
variation from 15 to 98 %, Nairobi station from 17 to 98 % and Kampala from 19 to 99 %. From Figure 4, it may
be seen that relative humidity variation over time is correctly captured by WRF for the Nairobi and Addis-Bole
stations. Despite this both the diurnal peaks and night lowest values seems to be not correctly reproduced by the
model that tends to overestimate the formers and underestimate the latter with a bias between -0.1 and 0.004 %.

However, WRF appears systematically to underestimate the relative humidity for the Kampala station showing a
mean negative bias. Different reasons could affect the underestimation of the relative humidity at this station. The
sensitivity of WRF model to the land use data (Teklay et al., 2019) connected with the proximity of Kampala to
Lake Victoria, which is a massive inland body of water (surface area 68,800 km$^2$) could influence the local
variation of relative humidity in ways which are not well reproduced by the model. The influence of Lake Victoria
and of the Kampala's complex topography on measurements of relative humidity was previously highlighted by
Singh et al. (2020) in relation to monthly visibility connected with PM levels. It has to be noted that relative
humidity was calculated from surface temperature and dew point values following Alduchov O. (1996) and not
directly sampled. A better agreement in the simulation of relative humidity from WRF can be found in the station
of Entebbe (n4 in Figure 2b) where the mean normalized bias shows a small underestimation of 0.04 %.

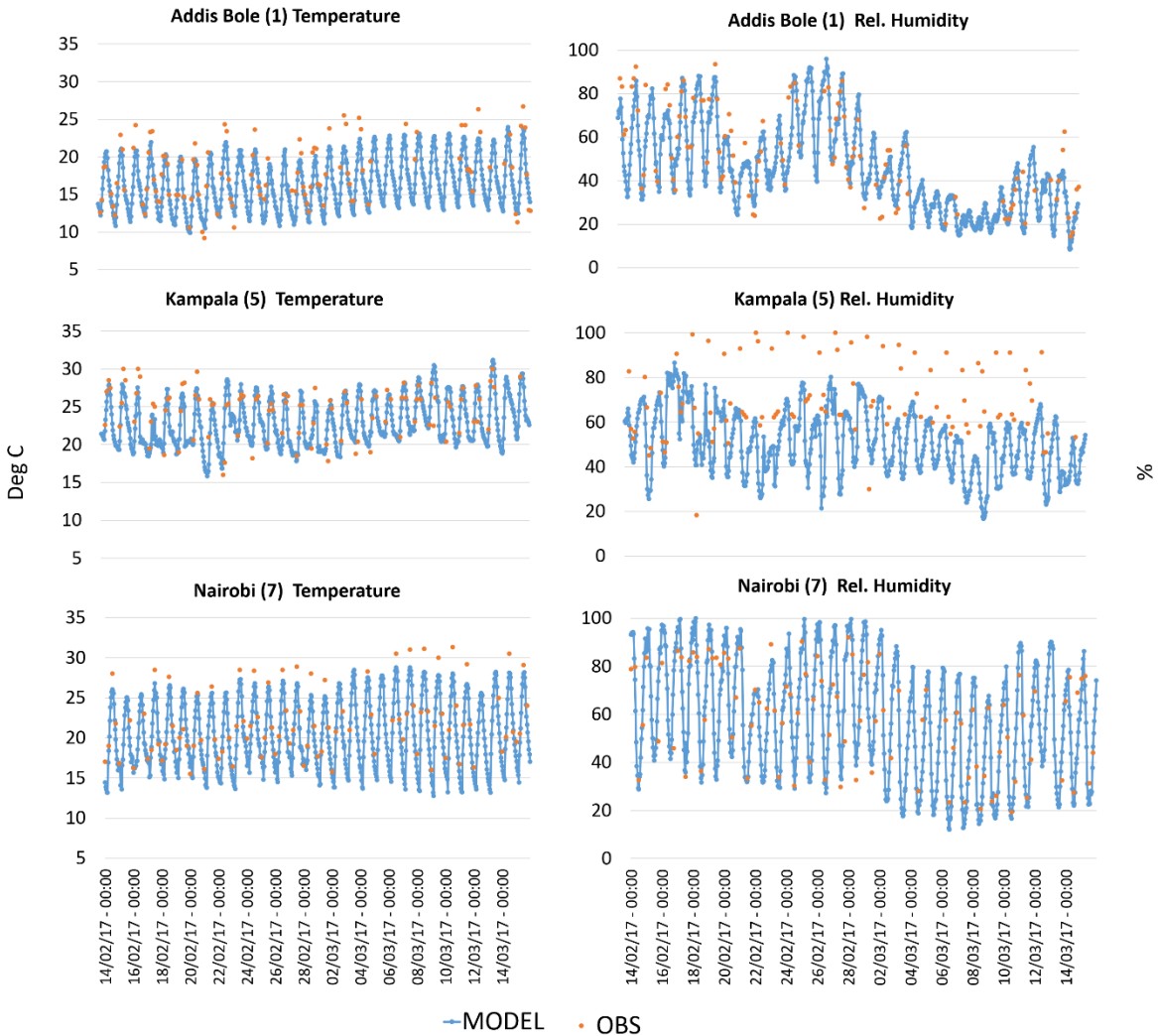

**Figure 4:** *Hourly time series of surface temperature (left column) and relative humidity (right column) for the closest ground weather stations to the urban centres of the cities of Addis Ababa (station 1 in Figure 2a), Kampala (station 5 in Figure 2b) and Nairobi (station 7 in Figure 2c). Comparison between modelled values (blue lines) obtained from the 2×2km domains and hourly observations (orange spots) from Met Office MIDAS database.*

Wind speed and directions from the urban stations of Addis Bole (n1 in Figure 2a), Kampala Station (n5 in Figure 2b) and Nairobi (n7 in Figure 2c) are shown in Figure 5 in the form of wind roses. WRF can reproduce the average wind directions in close agreement with the observed data for the analysed period for Nairobi showing the predominance presence of North-North-Eastern winds with high speed (> 4.0 m s$^{-1}$). Wind speed observations from the ground weather station of Kampala also suggest a strong southern wind component (> 4.0 m s$^{-1}$) while the model seems to reproduce a similar magnitude of the wind speed but on a larger range of directions ranging from the South-South-East direction to South-South-West. For Addis Ababa, WRF seems able to capture and reproduce the main wind directions observed for the simulated period, e.g., Eastern and North-Eastern winds. Despite this, slower winds between 0.2 and 2.0 m s$^{-1}$ with a strong North-Northeast component do not seem to be replicated by the model for the station located inside the capital of Ethiopia.

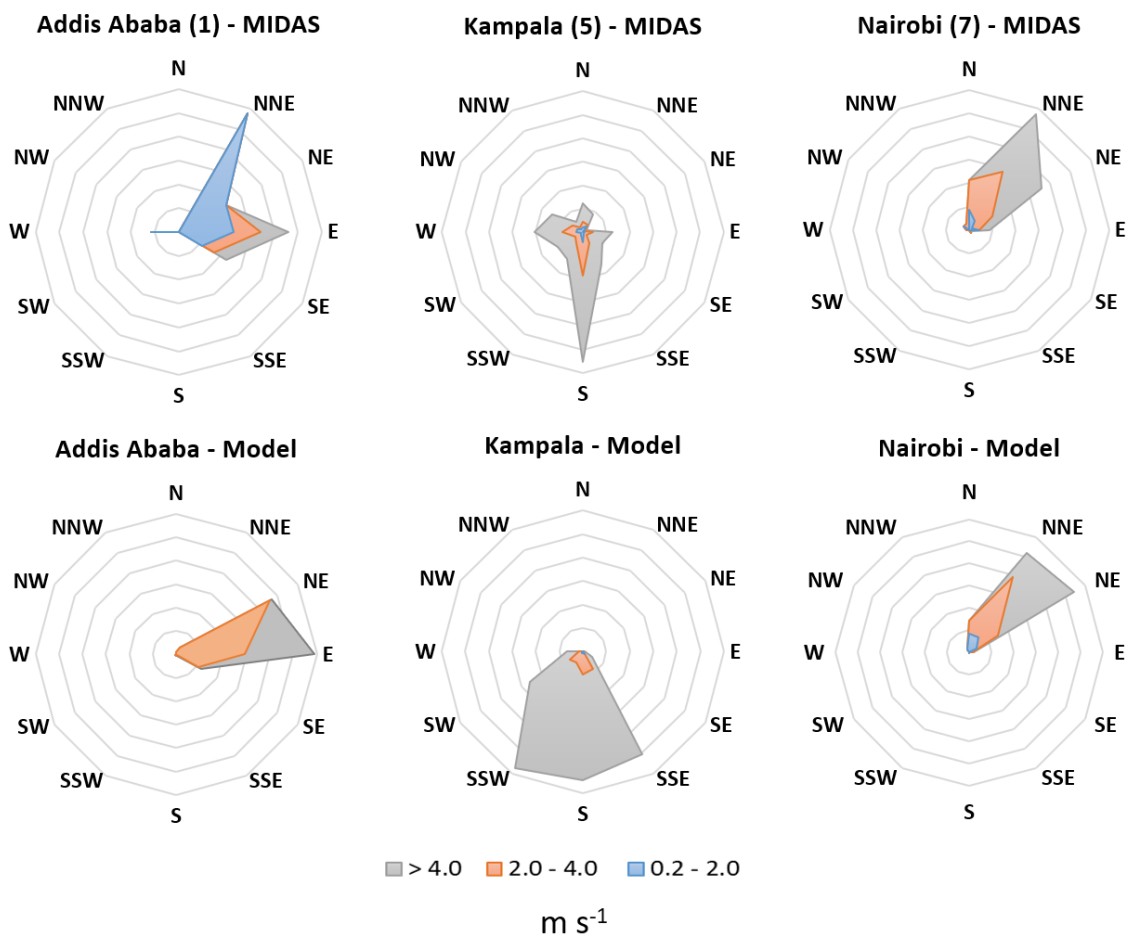

**Figure 5:** *Averaged wind roses for the whole analysed period (14th of February to 14th of March 2017) from the closest ground weather stations to the urban centres of Nairobi (n7 in Figure 2c), Kampala (n5 in Figure 2b) and Addis Ababa (n1 in Figure 2a) (MIDAS, top) and from WRF simulation outputs (Model, bottom).*

The lower agreement in the reproduction of the wind speed and direction in Addis Bole and Kampala stations can be connected to the particular locations of both stations. The difference in the location of the observations can, in fact, influence rapid changes in directions and speed locally recorded and not reproduced by the model. In the case of Kampala, the airport "Entebbe" is located near the coast of the Lake Victoria where the local conditions of wind are more susceptible of variation and can be erroneously reproduced by the model. In the case of Addis Bole, the only station settled in the urban area, the urban topography and possible canyon effects of the wind can be not well captured by the model that reproduces a more constant range of wind speed and directions not accounting for quick variations at low speed observed at the station.

The results obtained from the validation of the meteorological simulations performed over East African domains using WRF show that the model is on average able to reproduce all four variables taken in account close to the observed data in the 2×2 km domains with variable agreement between the three cities. The highest agreement in the weather analysis has been found for surface temperature with similar biases to Kerandi et al. (2017) and

relative humidity similar to Pohl et al. (2011), which is sufficiently accurate to be able to use these values for the physical calculations done by the chemistry transport model.

Nevertheless, the more detailed analysis of the urban weather stations revealed discrepancies in the reproduction of relative humidity and wind direction for the station of Kampala (UGA2K) that could affect the deposition, removal and transport processes simulated by CHIMERE and will be object of future investigation to further improve the meteorological performance of WRF. Even if the bias found for some variable in the calculation of the averaged statistics over all stations was high, the individual weather stations close to the urban areas of interest showed smaller bias and levels of MFB and MFE inside the goal or criteria range of performance and therefore considered acceptable for simulations.

**3.2 Validation of CHIMERE simulations**

The CHIMERE validation has been focused on the hourly levels of $PM_{2.5}$ modelled at the two observation sites for the domain KEN2K, representative of a roadside site and a rural background site. Also, from the urban background observational sites of the U.S. Embassies of Kampala (UGA2K) and Addis Ababa (ETH2K). The performance of CHIMERE was analysed also in terms of mean fractional error (MFE), mean fractional bias (MFB) and Pearson's coefficient (R) against the different level of average concentrations of $PM_{2.5}$ in the four observation points to evaluate the response of the model in reproducing low and high levels of hourly concentrations in comparison with observed values.

The validation of CHIMERE was done for the domains at highest resolution (2×2 km) despite the availability of emissions at a similar spatial resolution. The reason of this choice is motivated by the necessity to validate the reliability of the model against observation data from particular locations in different backgrounds. In order to better configure the model to represent the different urban and rural environments it is necessary to take in account the uncertainties of a model representation against an observation point. One cause of uncertainty when comparing modelling outputs with observations is the difference between a point measurement and a volumetric grid cell averaged modelled concentration (Seinfeld, 2016). On one hand, the extent of a measurement point, in fact, represents only the extent of the nearby points or an average concentration in a specified area. On the other hand, a surface level modelling grid typically has highest resolution of 1 km with a vertical height of between 20 and 40 m and the concentration represented by the model is the average over the entire grid cell.

In the particular case of the domains of East Africa, CHIMERE simulates at coarse resolution e.g., the 6×6 km, values of concentration representative of an average of 36 $km^2$, difficult to be compared with observations taken in a particular point. Increasing the spatial resolution and bringing it to 2×2 km the average value inside each grid cell will be representative of a smaller area such as 4 $km^2$ whose average value can be closer compared with an individual observation point.

*3.2.1 Statistical evaluation of model performances*

The absolute bias between mean observed and modelled concentrations of $PM_{2.5}$ shows an overestimation of the model for the domain KEN2K by between 0.01 and 3.7 µg m$^{-3}$ for Nanyuki and Nairobi, respectively, and for Addis Ababa (0.6 µg m$^{-3}$). On the contrary, the model underestimates $PM_{2.5}$ for the domain UGA2K (Kampala) by 7.2 µg m$^{-3}$ (Table 4).

The mean fractional bias (MFB) and error (MFE) for the two Kenyan observation points were found in both cases inside the goal performance criteria with MFE $\leq$ 50% and MFB $\leq \pm$ 30% both in Nairobi (roadside site) and in Nanyuki (rural site). The hourly MFB and MFE were 4.88 and 25.39 for Nairobi and 3.36 and 8.33 for Nanyuki while 0.1 and 1.99 for Nairobi and 1.08 and 4.73 for Nanyuki were the respective values found for the daily analysis.

The MFB and MFE analysis for the urban background site in Addis Ababa showed values inside the range of the goal criteria both for the hourly (2.93 and 29.99 for MFB and MFE) and for daily analysis (8.23 and 2.86). Finally, in the urban background site of Kampala the MFB were found inside the goal criteria both for daily (-11.28) and hourly (-7.60) analysis, while for the MFE the hourly analysis showed a value in the range of the criteria range (32.99) but daily MFE in the goal performance range (22.06) (Table 4).

The highest Pearson's coefficients (R) were found in Nanyuki with hourly and daily values of between 0.91 and 0.93. The roadside site of Tom Mboya Street in Nairobi had R values of between 0.35 and 0.38 while the urban background sites of Addis Ababa and Kampala had a lower agreement an hourly level (R values were between 0.10 and 0.29, respectively) than at a daily level (R values of between 0.42 and 0.30, respectively).

In general, the statistical analysis demonstrates that the model can reproduce the daily pattern of the hourly changes in concentrations for the two pollutants both in the three urban/roadside sites and in the rural site considered. The low R coefficient values obtained for the urban domains at the hourly level suggests that sources of anthropogenic emissions affecting urban air quality are still missing from the current emission inventory. Further work will be focused on the improvement of the magnitude of the emissions to better match the observed levels of concentrations of particulate matter at the urban level. Despite this and considering the daily average concentrations in the urban sites, the R coefficients were found to be between 30 and 42 % suggesting that CHIMERE better reproduces the concentrations of $PM_{2.5}$ using daily than hourly values.

The performance of CHIMERE varies between the domains of Kenya, Uganda, and Ethiopia. The performance of the model has been optimised during the validation for the simulation of hourly concentrations of $PM_{2.5}$ in Kenya and the same configuration applied to the domain of Uganda and Ethiopia to compare the reliability of the model. The difference in performance can be connected to different reasons: In first place, the difference in the sampling methods used for the two sites in Kenya against the measurements taken in the U.S. Embassies of Kampala and Addis Ababa. Secondly, another element of differentiation can be connected to the location of the observation sites in the cases of the U.S. Embassies and/or the possible influence of local sources not accounted in the emission inventories.

**Table 4:** *Hourly and daily statistical evaluation of CHIMERE model performance for the cities of Nairobi against ASAP observed data and against U.S. Embassies data for the cities of Addis Ababa and Kampala.*

| ASAP Observations | NAIROBI PM$_{2.5}$ (µg m$^{-3}$) roadside | | NANYUKI PM$_{2.5}$ (µg m$^{-3}$) rural | |
|---|---|---|---|---|
| | DAILY | HOURLY | DAILY | HOURLY |
| Model Mean | 58.3 | 58.3 | 3.24 | 3.24 |
| Observations Mean | 54.6 | 54.6 | 3.23 | 3.23 |
| MFB | 0.1 | 4.88 | 1.08 | 3.36 |
| MFE | 1.99 | 25.39 | 4.73 | 8.33 |
| R | 0.38 | 0.35 | 0.93 | 0.91 |
| U.S. EMBASSY Observations | A. ABABA – PM$_{2.5}$ (µg m$^{-3}$) urban | | KAMPALA – PM$_{2.5}$ (µg m$^{-3}$) urban | |
| | DAILY | HOURLY | DAILY | HOURLY |
| Model Mean | 18.7 | 18.7 | 36.2 | 36.2 |
| Observations Mean | 18.1 | 18.1 | 43.4 | 43.4 |
| MFB | 8.23 | 2.93 | -11.28 | -7.60 |
| MFE | 2.86 | 29.99 | 22.06 | 32.99 |
| R | 0.42 | 0.10 | 0.30 | 0.29 |

Finally, the site of Nanyuki is the location where the agreement between model and observations is highest. This site was chosen by Pope et al. (2018) as rural spot in a location of minimum local air pollution useful to calculate the net urban increment subtracting the rural background concentrations of Nanyuki from the urban concentrations in Nairobi. It is therefore intended by their work that the average concentrations in that site were really low. The model is able to reproduce this low level of contamination close to the reality and to reproduce also peaks of contamination in particular days of February probably generated elsewhere (see Section 3.2.2).

The MFB and MFE analysis have been conducted also at hourly level comparing modelling outputs and observations from all six sites in relation to the magnitude of hourly concentrations (Figure 6).

There are some MFB values outside the criteria range for PM$_{2.5}$ for the urban sites of Addis Ababa and Kampala and for the roadside site of Tom Mboya Street in Nairobi. In terms of the upper limit (MFB > 60 %) these values tend to be concentrated between 60 and 130 µg m$^{-3}$ for Tom Mboya Street, 40 and 55 µg m$^{-3}$ for Kampala and between 13 and 59 µg m$^{-3}$ for Addis Ababa (Figure 6). A much smaller number of MFB values for the Addis Ababa and Kampala sites are less than the lower criteria limit and these tend to be for lower concentrations between 10 and 26 µg m$^{-3}$.

MFE values outside the ranges of criteria are between 42-55 and 80-130 µg m$^{-3}$ for Tom Mboya Street, 43 and 60 µg m$^{-3}$ for Kampala and 13 and 59 µg m$^{-3}$ for Addis Ababa (Figure 6). The latter two sites present a more variability of MFB and MFE in comparison with the two sites of Kenya where is visible a common positive bias of the model in reproducing the highest concentration levels. The reliability of the model is therefore higher for the domain of Kenya, both for a rural and for a roadside site than for the two urban background sites in Uganda and Ethiopia.

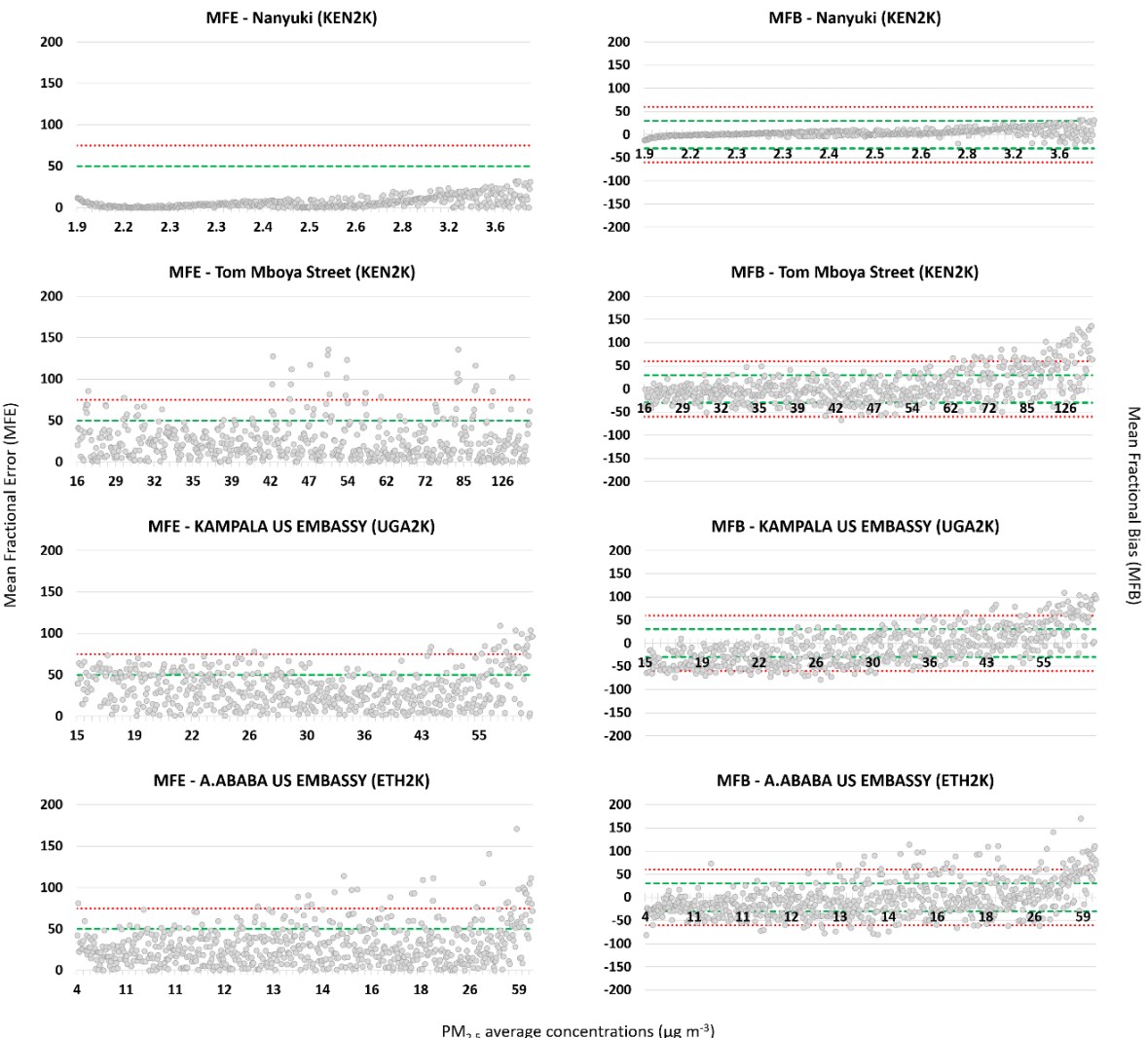

629

**Figure 6:** *Hourly mean fractional bias (MFB) and mean fractional error (MFE) values calculated for the locations of Tom Mboya Street and Nanyuki (KEN2K), Kampala U.S. Embassy (UGA2K) and Addis Ababa U.S. Embassy (ETH2K) for the analysed period against hourly concentrations of PM$_{2.5}$. The green lines represent the MFB range ±30 % and the MFE limit of 50 % for which the model performance can considered reliable, the red lines represent the MFB range ±60 % and the MFE limit of 75 % for which model performance can be increased by diagnostic analysis on the chemical precursors of PM$_{2.5}$.*

The overall performance of the model against different levels of concentrations is summarised in Table 5. The PM$_{2.5}$ reproduced at the two sites in KEN2K shows a higher percentage of values within the MFB and MFE performance goals for the rural site of Nanyuki, than for Tom Mboya Street. e.g., 97 % compared to 69 % and 99 % compared to 88 % for the MFB and MFE measures respectively. For the criteria measure, the corresponding percentages are 2 % vs. 22 % and 1 vs. 7 % (Table 5).

The percentages for the urban sites of Kampala and Addis Ababa show a lower agreement between the model and observations. For the former 48 % of the values according to the MFB measure are within the goal range, 37 % are within the criteria range and 15 % are outside. For the latter, according to the MFB criteria, 57 % of the values are inside the goal range, 30 % of values are within the criteria range and 13 % are outside. In terms of the MFE measure, 74 % and 80 % of values for the two cities are within the goal range, 16 % and 11 % within the criteria range and 10 % and 9 % outside respectively (Table 5).

648

**Table 5:** *Hourly mean fractional bias (MFB) and error (MFE) percentage of points inside the goal limit (GOAL), inside the diagnostic range (CRITERIA) and outside the reliability criteria (OUT) from model outputs extracted from the four analysed locations.*

| Location | MFB | | | MFE | | |
|---|---|---|---|---|---|---|
| | GOAL (%) | CRITERIA (%) | OUT (%) | GOAL (%) | CRITERIA (%) | OUT (%) |
| **Tom Mboya Street (KEN2K)** | 69 | 22 | 9 | 88 | 7 | 5 |
| **Nanyuki (KEN2K)** | 97 | 2 | 1 | 99 | 1 | 0 |
| **Kampala (UGA2K)** | 48 | 37 | 15 | 74 | 16 | 10 |
| **A. Ababa (ETH2K)** | 57 | 30 | 13 | 80 | 11 | 9 |

According to the methodology proposed by (Boylan and Russel, 2006) the performance of a modelling system is fairly good for $PM_{2.5}$ representation if about the 50 % of the points are within the goal range and a large majority are within the criteria range. From the analysis of the four sampling sites the values of MFB inside both the goal and range for Tom Mboya Street are 69 %, 97 % for Nanyuki and 57 % for Addis Ababa and only for Kampala are 48 %. Similarly, for the MFE measure, 99 % for Nanyuki, 88 % for Tom Mboya Street, 80 % for Addis Ababa and 74 % for Kampala are inside both the goal range. The demonstrates that the performance of the model can be considered to be satisfactory (Table 5).

Finally, the reason for the presence in the Addis Ababa and Kampala simulations of values outside the criteria range both at high and at low concentrations of $PM_{2.5}$ can be connected to the representation of the original PM emissions in the combined inventory. It is possible that CHIMERE is not able to correctly reproduce all the chemical processes involved in the secondary formation of inorganic and organic individual components of $PM_{2.5}$ with the extent of the present input data. Moreover, the possible misrepresentation of local emission sources not reproduced in DICE-EDGAR can also affect the performance of the model. Finally, the different location of the urban background observation sites and the sampling techniques for PM observation can also have a key role in the correct detection of the concentrations.

*3.2.2 Hourly variation of $PM_{2.5}$ in urban and rural sites of Kenya*

Hourly modelled variation of $PM_{2.5}$ levels obtained by CHIMERE compared with observations are shown for the urban sampling site of Tom Mboya Street in Nairobi and for the rural site of Nanyuki (Figure 2c).

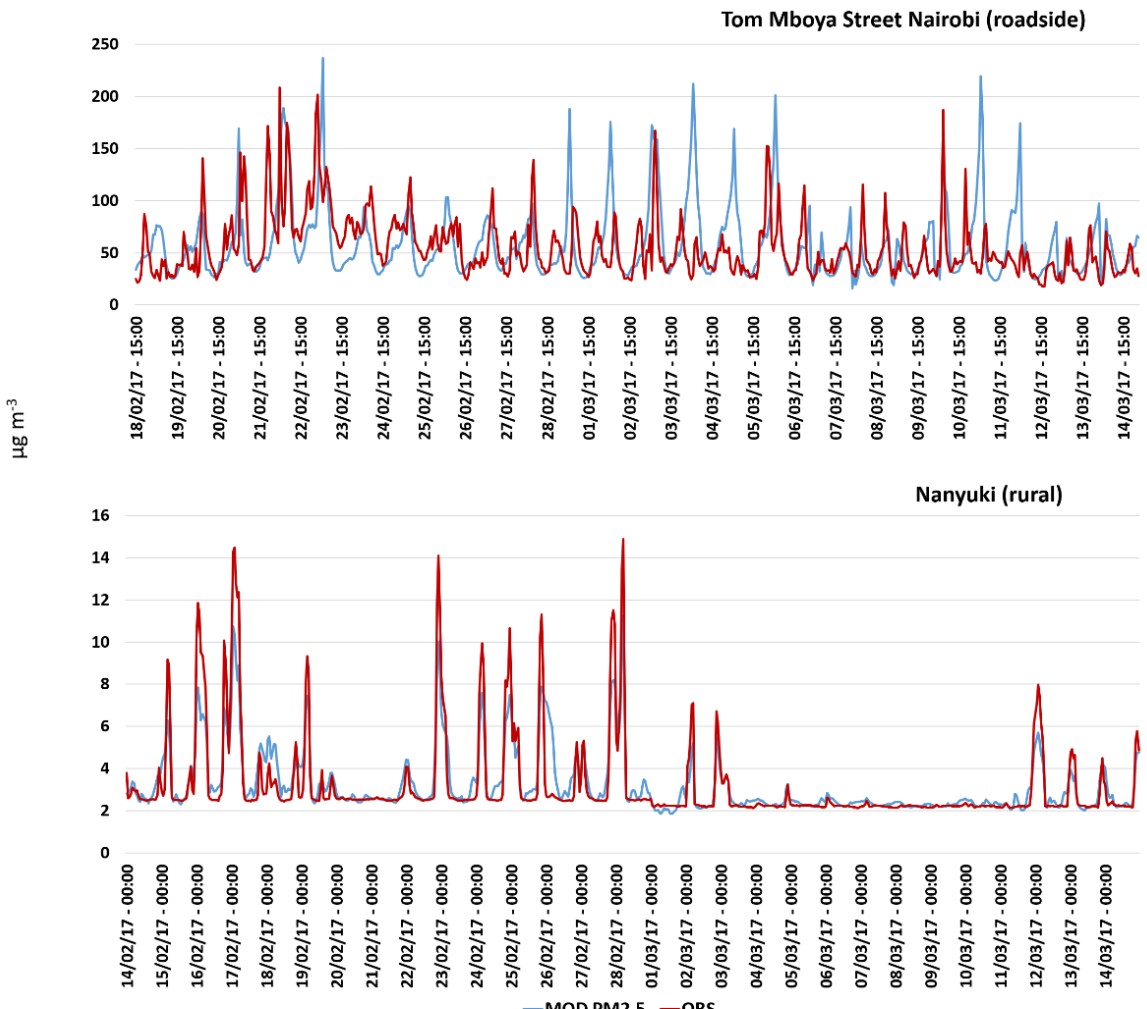

**Figure 7:** *Hourly time series for PM₂.₅ from the roadside of Tom Mboya Street (top) and from the rural site of Nanyuki (bottom) from modelled output from CHIMERE model (blue line) and observed values from Pope et al. (2018) (red line) for the analysed period The simulation started on the 14th of February. For the Tom Mboya Street site only the period of time between the 18th of February and the 14th of March when observations were available has been shown in the timeseries.*

By inspection of Figure 7 it can be seen that CHIMERE is able, in general, to reproduce the daily variation of PM₂.₅ across the simulated period at both sites. The magnitude of the emissions adopted seems to be suitable both for the roadside area of Tom Mboya Street and for the rural background site of Nanyuki, with higher agreement shown by the latter. CHIMERE captures only part of the daily peak observed in Tom Mboya Street with comparable magnitude but misrepresents some peaks. In particular it models higher hourly peaks than those observed as previously mentioned in the MFB and MFE analysis.

The misrepresentation of some high peaks in Tom Mboya Street is possibly due to a number of different reasons. Firstly, is important to recall that the point measurements and relative observed concentrations are representative of a smaller portion of space in comparison with grid-cell concentrations modelled. In this particular case the comparison is between a roadside site subjected to possible additional local sources of PM₂.₅ not accounted for in

the emissions and not correctly reproduced by CHIMERE. On the other hand, a few of the modelled peaks were
overestimated. This can be addressed by improved temporal description of the emissions and in their magnitude
in comparison to the reality. As mentioned previously, the anthropogenic emissions used in this work were the
most up-to-date available at the time and that there is inevitably some difference between the measured data due
to the difference in time between the inventories and the measurements. Despite this, there is reasonable agreement
between model outputs and observed concentrations for the majority of the analysed period highlighting the
reliability of CHIMERE in describing the hourly concentrations trends for a roadside site with expected high
levels of $PM_{2.5}$ contamination.
Similarly, in the rural site of Nanyuki, the model seems to correctly reproduce the hourly variation of the
concentrations during the whole period, underestimating the maximum peaks at the beginning of February and in
the last four days of simulation in March. (Figure 7). The site shows different magnitude in the concentrations of
$PM_{2.5}$ when comparing the February and March periods. While between the 4[th] and the 10[th] of March hourly
concentrations are around 3-4 µg m$^{-3}$, previously and subsequently to this period of time, the concentrations of
$PM_{2.5}$ are more than two times higher. This behaviour is visible both in the observations from the site (red line in
Figure 7, bottom) and from the model outputs obtained using CHIMERE (blue line in Figure 7, bottom).
The site of Nanyuki was chosen by Pope et al. (2018) as rural spot in a location of minimum local air pollution
influence. Data from Nanyuki was used for the calculation of the net urban increment subtracting the rural
background concentrations of Nanyuki from the urban concentrations in Nairobi. The average concentrations
around 3-4 µg m$^{-3}$ in the period between the 4[th] and the 10[th] are, on one hand, levels of the rural background in
absence of any external influence from meteorological parameters and in absence of local sources.
On the other hand, the presence of higher hourly peaks in before and after the 4[th] to 10[th] can be linked to different
reasons: the presence of local emission sources contributing to the peaks or the dispersion of polluted air masses
from elsewhere towards the site of Nanyuki. It is important to observe that model and observations seems to agree
particularly well in the description of the difference in magnitude between the different time periods excluding
the possibility that the observed values can be influenced by local emission sources not accounted in the emission
inventory. It seems more likely that those concentration levels are transported to Nanyuki from neighbouring areas
with higher levels of $PM_{2.5}$ contamination. To investigate this possible role of $PM_{2.5}$ dispersion towards Nanyuki,
we consider the closest MIDAS weather station to the sampling area of Nanyuki, in the town of Nyeri (0.43°S,
36.95°E altitude 1916 m a.g.l.) (n10 in Figure 2). Nyeri is only 60 km from the Nanyuki site and is situated
between Mount Kenya (0.10°S, 37.30°E, altitude 4341 m a.g.l.) to the west and the Aberdare Range (0.46°S,
36.69°E, altitude 3441 m a.g.l.).
The daily average concentrations observed in the sampling site of Nanyuki have been compared with the daily
mean values of wind speed and directions observed at the MIDAS station of Nyeri and with the daily mean values
of wind speed and directions modelled by WRF in Nanyuki (Figure 8). The period between the 4[th] and the 10[th] of
March, when the daily average concentrations of $PM_{2.5}$ observed in Nanyuki were around 2.2 µg m$^{-3}$ corresponds
to higher wind speed conditions (between 4 and 5 m s$^{-1}$) mainly coming from North-Est (around 60 degrees). In
the same period, at Nyeri the modelled wind speed was low (between 1 and 2.5 m s$^{-1}$) and mainly with a westerly
component (between 220 and 300 degrees).

In the periods of higher average daily concentrations of PM$_{2.5}$ between the 15$^{th}$ and the 19$^{th}$ and between 22$^{nd}$ and
the 28$^{th}$ of February 2017, both in Nyeri (using observations) and in Nanyuki (using model outputs) the component
of wind directions seems to be consistent in reproducing southern winds (between 120 and 190 degrees) with wind
speeds between 1.5 and 2.5 m s$^{-1}$ in the first period and between 2 and 3 m s$^{-1}$ in the second period.

The correspondence between the wind speed and directions in particular time periods and the vicinity of the towns
could suggest the potential dispersion of pollutants from the southern area where the hotspot of Nyeri is located
upwind in the northern area of Nanyuki (downwind) in accordance with the wind fluxes from south to north from
Nyeri from the observations and also from WRF outputs extracted from the Nanyuki location. The flux could also
be driven by the location of Nyeri sited at the entrance of a basin between two mountain ranges. On the other
hand, in the period of low concentrations between the 4$^{th}$ and the 10$^{th}$ of March north-eastern winds (around 60
degrees) blow with high speed on Nanyuki (around 4 m s$^{-1}$) while lower speed winds (between 1 and 2 m s$^{-1}$) from
a more variable directions (between 170 and 300 degrees) are blow in Nyeri preventing the possible dispersion of
pollutants.

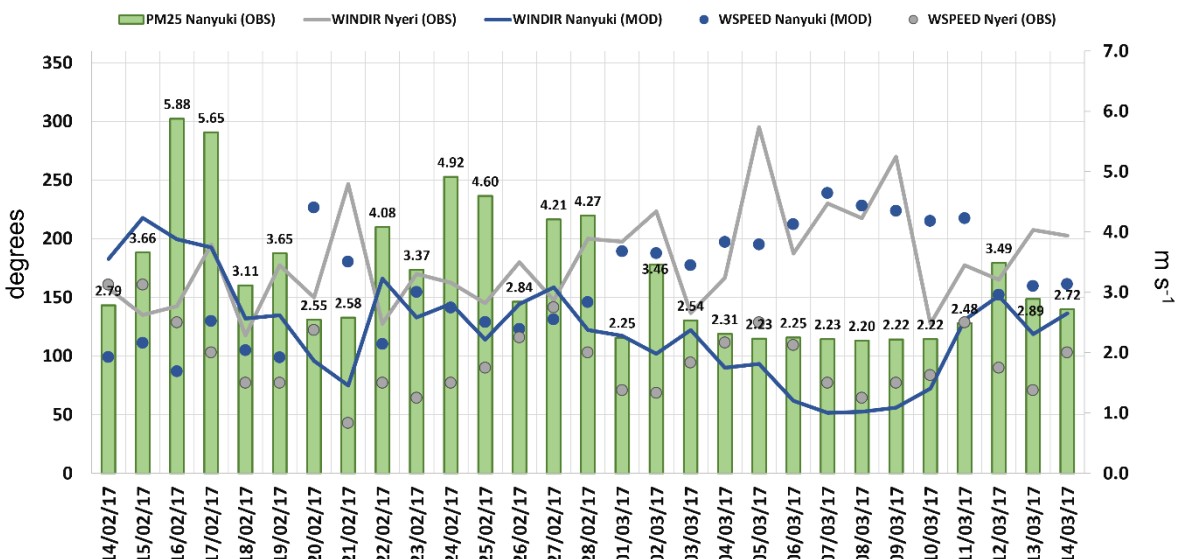


**Figure 8:** *Comparison between daily observed values of wind speed (grey spots) directions (grey lines) from the MIDAS site*
*of Nyeri (n10 in Figure 2c), modelled daily wind speed (blue dots) and directions (blue lines) from the site of Nanyuki with*
*daily average observations of PM$_{2.5}$ (expressed in μg m$^{-3}$, green columns) obtained from the sampling site of Nanyuki (red dot*
*in Figure 2c).*

The present analysis was done on the relationships between weather conditions and the relative correspondence
in hourly and daily levels of PM$_{2.5}$. Further analyses are necessary to clarify the possible presence of additional or
alternative factors influencing the changes in concentrations observed and modelled by CHIMERE. The presence
of possible precipitations during the low concentration period could represent an alternative possibility the change
in concentrations. Despite this no precipitation were recorded during that period according to Pope et al. (2018)
and no precipitation was modelled by WRF in that time period. Nevertheless, the lack of additional weather
observations in the sampling site of Nanyuki and middle way between the two towns prevent from any additional
hypothesis in relation to the presence of possible pollutant transport phenomena that will be object of future
investigations. Further efforts will be oriented in a more detailed trajectory analysis of the winds and in a more
detailed representation of the emissive sources present in the area to investigate possible transport effects in this
area.
The average concentrations of $PM_{2.5}$ for the entire period of simulation between the 14[th] of February and 14[th] of
March 2017 are shown for the domain centred over Kenya with spatial resolution of 2×2 km (KEN2K, Figure 9).
Highest average concentrations during the monthly period are modelled in the urban area of Nairobi (defined by
the red dashed square in Figure 9) with highest average values inside the city around 80 µg m$^{-3}$. The concentrations
are spread on average in the southwest area of the city and on the northeast side in direction of the conurbation of
Thika and Makuyu. These towns became part of the Metropolitan Area of Nairobi in 2008 due to the rapid increase
in population and urbanization of the area (UNEP, 2009) and represent a large hotspot of emissions of $PM_{2.5}$ with
concentrations modelled between 20 and 30 µg m$^{-3}$ as average of the entire period. Other hotspots of concentration
of $PM_{2.5}$ found in the domain are the city of Nakuru with average concentrations between 20 and 40 µg m$^{-3}$ and
the area between Nyeri, Embu, Meru and Siakago with average concentrations around 20 and 30 µg m$^{-3}$ (Figure
9). The average of the modelled concentrations in the area of Nanyuki is generally smaller, with concentration not
exceeding 10 µg m$^{-3}$ in the whole area.

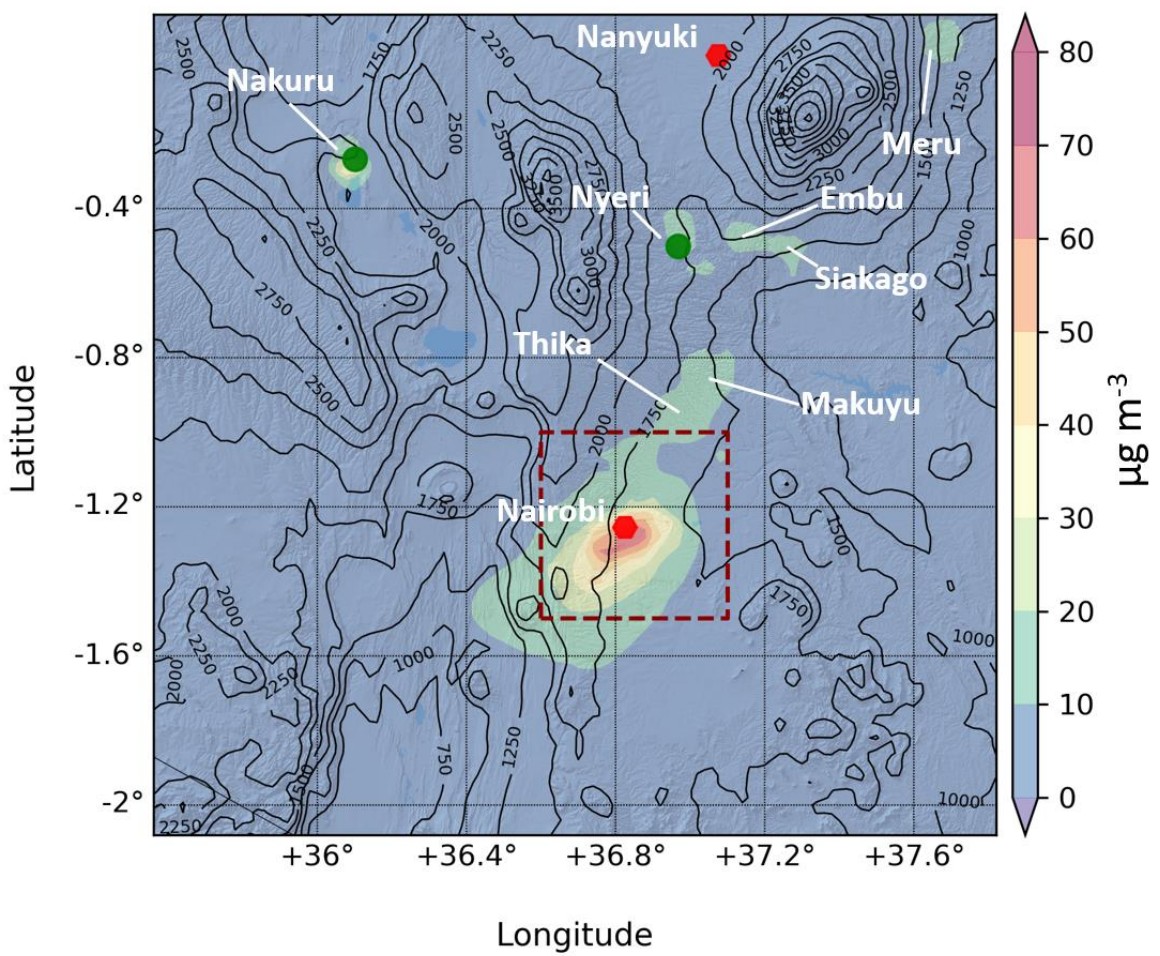


**Figure 9:** *Average concentration of PM$_{2.5}$ for the whole simulated period for the domain KEN2K at spatial resolution of 2×2*
*km. The map shows the location of the hotspots with higher average concentrations modelled by CHIMERE for the entire*
*period. The red dashed square shows the urban domain of Nairobi analysed for the Air Quality Indexes analysis in section*
*3.3.*

**3.3 CHIMERE as an Air Quality Management Tool**

The usefulness of CHIMERE as a decision support tool to facilitate air quality management of large urban
conurbations of SSEA was investigated for the three domains at a resolution of 2×2 km, namely: KEN2K, UGA2K
and ETH2K. Daily observations of PM$_{2.5}$ for the three domains were compared with modelled concentrations in
terms of number of exceedances from the WHO limit of 25 µg m$^{-3}$ observed and captured by the model (Figure
10). For the limited case of Nairobi, hourly average concentrations for the whole monitored period were compared
with Air Quality Indexes data and the spatial distribution of daily average concentrations on the constituencies
was analysed, highlighting how many areas of the city showed poor air quality indexes during the analysed period
(Figure 11).

Daily concentrations of PM$_{2.5}$ modelled by CHIMERE were compared with the number of exceedances of the
WHO limit of 25 µg m$^{-3}$ observed during the simulated period. Figure 10 shows the daily average concentrations
for the three cities in the sampling sites used for the validation of the model. It can be seen that Nairobi and
Kampala have the highest number of exceedances from the WHO limits (24) followed by Addis Ababa with only
6 observed exceedances. From Table 6 it can be seen that CHIMERE provides sufficient accuracy to detect the
exceedances of $PM_{2.5}$ from the WHO limits. In particular, it was able to detect 67 % of the exceedance for Addis
Ababa with only two false positives, 91 % for Kampala and all of the exceedances for Nairobi without any false
positives.

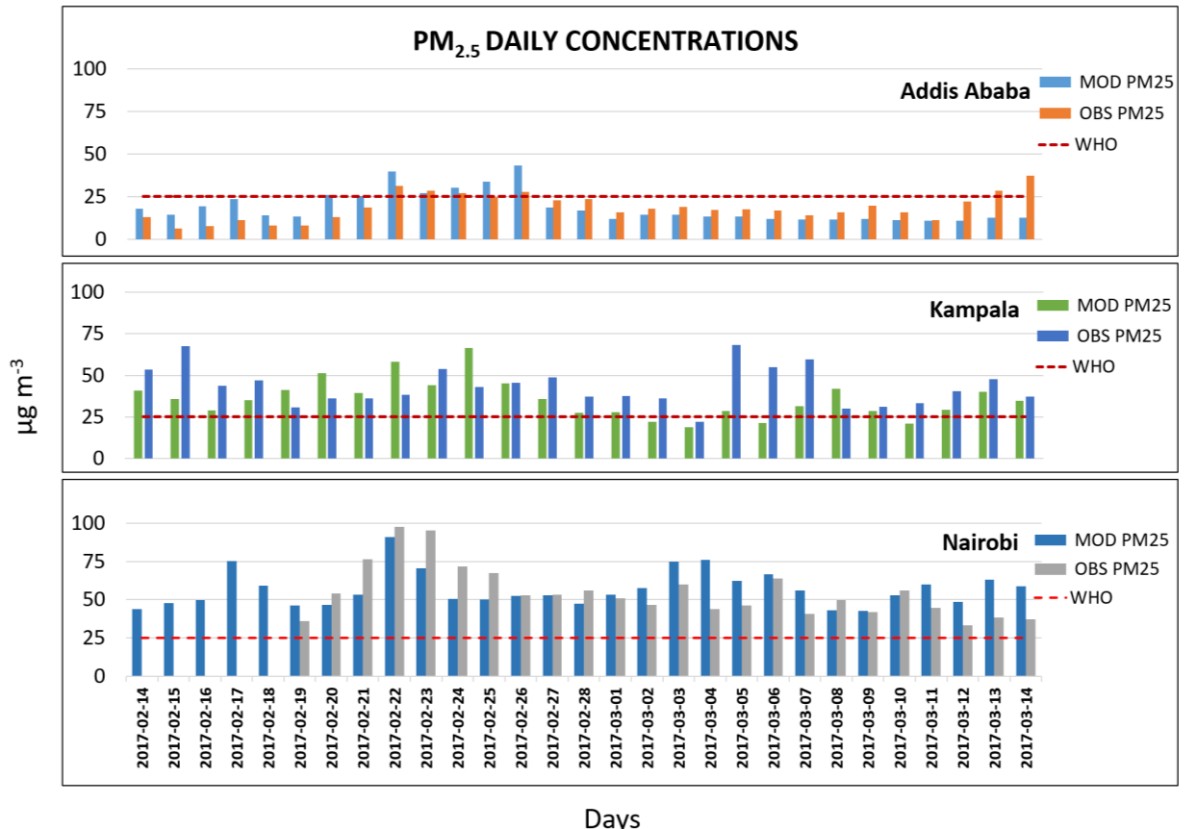


**Figure 10:** *Daily concentrations of $PM_{2.5}$ between the 14th of February and 14th of March obtained from CHIMERE outputs*
*from domains at 2×2 km compared with U.S. Embassy daily totals for the cities of Addis Ababa (top) and Kampala (middles)*
*and with ASAP observations for the city of Nairobi (bottom). All three simulations have been compared also with the WHO*
*threshold limit for $PM_{2.5}$ concentrations (red line). For the case of Nairobi, only observations from the 18th of February were*
*available.*
The Air Quality Index (AQI) represents the conversion of concentrations for fine particles such as $PM_{2.5}$ to a
number on a scale from 0 to 500 (Table 6). The higher the AQI value, the greater the level of air pollution and the
greater the health concern. AQI values at or below 100 are generally thought of as satisfactory. When AQI values
are above 100, air quality is unhealthy: at first for certain sensitive groups of people (101 – 150), then for everyone
as AQI values get higher (> 151) (EPA, 2012).

**Table 6:** *Summary of the number of WHO exceeding limits for $PM_{2.5}$ during the simulated period from the 14th of February to*
*the 14th of March 2017 observed and modelled.*

| Cities | Exceedances of WHO limits (observed) | Exceedances of WHO limits (modelled) |
|---|---|---|
| **Nairobi** | 24 | 24 |
| **Addis Ababa** | 6 | 4 |
| **Kampala** | 24 | 22 |


The daily average concentrations of $PM_{2.5}$ during the analysed period between the 14th of February and 14th of
March 2017 have been averaged for the urban area of Nairobi (red square in Figure 9 and Figure 11) and compared
with the city constituencies spatial extension according to data from the Open Africa dataset (Open-Africa, 2018).
According to the division, 17 are the constituencies inside the Nairobi city boundaries (Figure 11). Averaged daily
concentrations of $PM_{2.5}$ show that 8 of 17 constituencies had AQI values between 55.5-150.4 µg m$^{-3}$ during the
whole period. These areas are the most central and urbanized of Nairobi. Starehe constituency (n13 in Figure 11)
contains the Tom Mboya Street sampling site (black spot in Figure 11) previously discussed where the WHO
limits for $PM_{2.5}$ have been systematically exceeded during the analysed period. According to the SEDAC
population density data this area has population density between 15,000 and 30,000 people/km$^2$ exposed to AQI
between 151-200 corresponding to unhealthy category for human health. Finally, the Langata constituency
(magenta spot in Figure 11) has a population of 176,000 people and shows average levels of $PM_{2.5}$ of 45 µg m$^{-3}$,
unhealthy for sensitive groups of people.

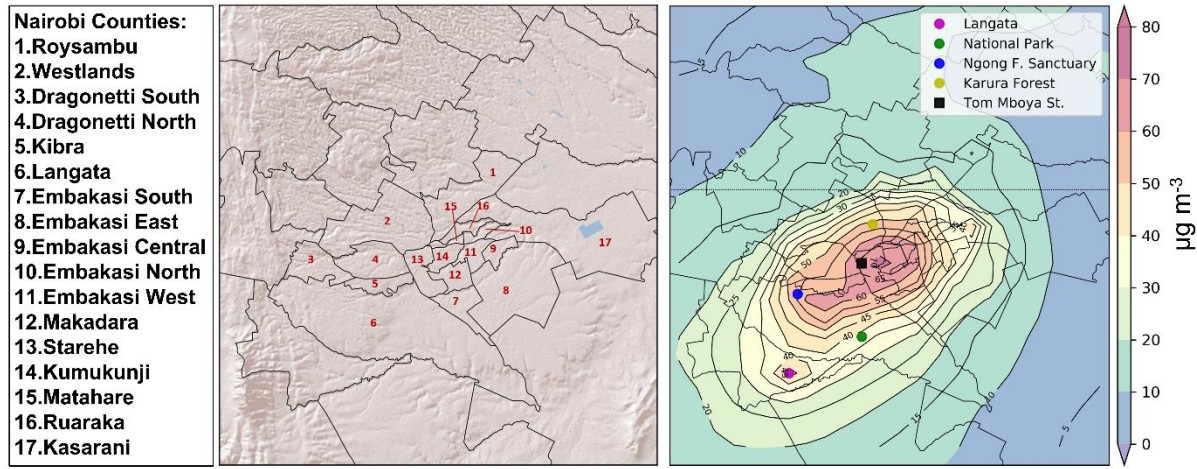


**Figure 11:** *Map showing the urban area of the city of Nairobi shown as dashed square in Figure 9. The constituency division*
*of Nairobi (left) from Open Africa dataset (Open Africa, 2018) is compared with the average hourly concentrations of $PM_{2.5}$*
*over the analysed period (right).*

Moreover, Nairobi has a number of natural areas on the outskirts of city. Some particular locations such as the
Karura Forest (yellow spot in Figure 11) and the Ngong Forest Sanctuary (blue spot in Figure 11) show averaged
daily levels of $PM_{2.5}$ around 50 and 55 µg m$^{-3}$ corresponding to an AQI of between 101 and 150 (e.g., unhealthy
for certain sensitive groups of people). According to SEDAC data, the population density is between 10,000 and
15,000 people/km$^2$ in this area. Similarly, in the south side, near the entrance to the Nairobi National Park (1.36°
S, 36.82° E, green spot in Figure 11) the average daily levels of $PM_{2.5}$ are approximately 40 µg m$^{-3}$ with AQI
values between 101 and 150 with a population density around 10,000 people/km$^2$. This area (surface area 117
km$^2$) has been impacted by a rapid urbanization since 1973 with a consequent increase of human activities
including settlement, pastoralism and agriculture (Ogega O.M., 2019). These activities have already made it
difficult for wildlife to migrate to and from the Nairobi National Park also are resulting in a deterioration of air
quality. The rapid increase of population density in the south side of Nairobi seriously risk increasing the level or
AQI exposing more people to harmful level of $PM_{2.5}$.

**4 Conclusions**



The WRF and CHIMERE models were configured and validated to simulate the air quality levels of PM in Eastern
Sub-Saharan African urban conurbations.

In order to obtain updated anthropogenic emissions for 2017, the global EDGAR inventory and the DICE
inventory for Africa were merged and spatially distributed using population density data for the year 2017
obtained by linear extrapolation.

WRF showed a variable capability in reproducing the main surface weather variables according to the different
conditions of the three domains. A lower agreement between observations and the model was observed in Kampala
for relative humidity and wind speed. The analysis was carried out on all surface meteorological stations available
from the MIDAS network on a three-hourly basis. A further meteorological analysis extended to vertical profiles
could reveal possible limitations of the model. However, the absence of vertical meteorological data limited the
analysis and validation to ground level only.

CHIMERE was able to reproduce the daily levels of $PM_{2.5}$ for the urban site of Nairobi as well as for the rural site
of Nanyuki. The 69 % of the MFB values and 88 % of the MFE value were inside the highest confidence area for
Nairobi and the 97 % and 99 % for Nanyuki attesting that the agreement between the observed and modelled data
was sufficient to allow for quantitative analyses of daily average concentrations. Similar findings were also found
for the other two urban background domains of Addis Ababa (57 % for MFB and 80 % for MFE) and Kampala
(48% for MFB and 74 % for MFE) despite different characteristics and sources of observation being used for the
validation. The discrepancies observed in the hourly trends of $PM_{2.5}$ modelled by CHIMERE compared to
observed values in the urban sites suggest that further studies are needed in the three urban areas. These studies
are required to improve the understanding of the typology and quantity of local emission sources, which are
sometimes misrepresented or absent in global emission inventories. This will enable the chemical processes acting
in the urban troposphere to be adequately characterised and thereby actual air quality levels to be determined.

Nevertheless, using existing data sets, CHIMERE has shown reliability in reproducing both hourly and daily levels
of $PM_{2.5}$ with hourly values largely inside the range of reliability connected with mean fractional bias and error.
The merged emission inventory DICE-EDGAR, despite the low resolution was able to return a correct magnitude
for the emissions in representation of urban and rural context. Despite this, few urban peaks observed in Nairobi
have been missed by CHIMERE or in other cases misrepresented highlighting the necessity of further efforts in
the creation of newer emission inventories for SSEA. In the light of this, the possibility to develop local emission
inventories, ideally at high spatial resolution it would represent a significant step ahead in the air quality research
in this area of the world. Despite this and at the extent of the present data, CHIMERE showed enough robustness
and reliability to be adopted as a decision support tool for the management of air quality, correctly reproducing
most of the exceedances of the limits set by the WHO for $PM_{2.5}$ for all three cities considered.

The analysis focused on the average concentrations of $PM_{2.5}$ for the domain of Kenya revealed that the
metropolitan area of Nairobi represents a big hotspot of air pollution but that also small cities located in the
outskirts of the capital of Kenya showed worrying levels of atmospheric contamination. These levels of air
pollution have the potential capability to affect also rural areas where the local emissions are rare or not present.
The possibility of transport phenomena of $PM_{2.5}$ towards these areas, however, is still to be verified. The work
has also shown for urban area of Nairobi the presence of low and unhealthy air quality indexes in 8 of 17 its
constituencies and the relative population density exposed to harmful level of air contamination. Moreover, a
number of natural areas in the outskirts of Nairobi have similarly low levels of AQI and increasing population
highlighting how the problem of poor urban air quality due to rapid urbanisation, anthropogenic activities and
lack of regulation can also detrimentally affect and deteriorate natural habitats.
The present work represents a first step in the use of numerical models for atmospheric chemistry simulations in
East Africa with particular focus on urban conurbation. The aim of the present work was to assess the possibility
to perform simulations with results close to observations in order to open the road for more detailed works. The
natural next step of the present research aims to refine the quantity and quality of the input data used for the
validation of both modelling system in order to improve the reliability of the predictions. Moreover, a more
detailed analysis of the secondary inorganic and organic components of $PM_{2.5}$ will be conducted for the three
domains. Finally, the performance of CHIMERE will be tested in the reproduction of gaseous species too in order
to give a wider vision of the capabilities and opportunities of numerical modelling in this area of the world with
present data. Additional future efforts to improve the calibration and validation of the modelling system, especially
relating to meteorology, will focus on assessing the dispersion dynamics of contaminants through urban centres
and possible pollution transport events from urban to rural areas. To aid this, further work is required by local
East African authorities and research bodies to improve the quantity and the quality of data for weather and air
quality simulations. However, in this work, we have shown that currently available data is sufficient to carry out
simulations of air quality that can be used for quantitative evaluation of anthropogenic emissions impact and to
support mitigation policies at the local level.
**Authors Contribution: Andrea Mazzeo:** Conceptualization, Methodology, Software, Validation, Writing-
Original draft preparation, Writing- Reviewing and Editing. **Michael Burrow:** Supervision, Writing - Review &
Editing **Andrew Quinn:** Supervision, Resources. **Eloise A. Marais**: Data curation, Resources, Writing - Review
and Editing. **Ajit Singh**: Resources, **David N'gang'a:** Resources, **Michael Gatari:** Resources. **Francis Pope**:
Supervision, Data curation, Funding acquisition, Writing - Review and Editing.
**Acknowledgements:**
The work is funded by the UK Department for International Development (DFID) via the EPSRC grant 'Digital
Air Quality' (EP/T030100/1) and by the East Africa Research Fund (EARF) grant 'A Systems Approach to Air
Pollution (ASAP) East Africa'. The practical support of the Schools of Geography and Earth Sciences and
Engineering at the University of Birmingham are gratefully acknowledged.
**Data Availability:** the combined DICE-EDGAR anthropogenic emission inventory is downloadable from:
https://doi.org/10.25500/edata.bham.00000695. CHIMERE model is downloadable from:
https://www.lmd.polytechnique.fr/chimere/ while WRF model is downloadable from:
https://www2.mmm.ucar.edu/wrf/users/download/get_sources.html, Weather observations used for the validation
of WRF have been downloaded from the Met Office:
http://catalogue.ceda.ac.uk/uuid/220a65615218d5c9cc9e4785a3234bd0. Data relative to observations of PM$_{2.5}$
for Nairobi (Kenya) are available upon request to the authors of Pope et al. (2018) while observations of PM$_{2.5}$
for Addis Ababa (Ethiopia) and Kampala (Uganda) are available upon request to the respective U.S. Embassies.

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
