# Peer review of "Evaluation of WRF and CHIMERE models for the simulation of PM2.5 in large East African urban conurbations."

_Atmospheric Chemistry and Physics, 2021_

## Author Comment (AC1)

*General Comments:*

*Overall, this paper uses a sound scientific approach that supports its main conclusion that the coupled WRF-CHIMERE model is suitable as a tool for air quality management in East Africa. It could be improved with more in-depth discussion of some results and clarification/correction of several minor items.*

***Specific Comments:***

*Q. I couldn't tell from the text if anthropogenic emissions inputs to the model had any diurnal variations. Only annual total emissions are mentioned, possibly indicating that constant emission rates are used for each pollutant. This could have a major impact on correlation to hourly data.*

A > The anthropogenic emission inputs have been temporally disaggregated using factors included in the CHIMERE pre-processor "emisurf2016". These coefficients are divided by emission sector and permit to save the total mass of the original emissions (provided in the emission inventories) but to divide this total according to the diurnal variation. This system, largely used in atmospheric chemistry modelling, permit to allocate e.g., more emissions from road transport in the peak of the morning and afternoon rush hours than in the other hours of the day preserving the original total value of the emissions. A New line highlighting this aspect has been added at **233-234**.

*Q. Not entirely clear how statistical measures are averaged (lines 336-338). Are measures calculated for each site and then those values for each domain are averaged (e.g., the 5 Relative Humidity NRMSE values for the 5 KEN2K weather stations are averaged to produce the KEN2K NRMSE value?) Or are the observed and modelled data for all the sites within a domain used together to calculate the average measure?*

A > The statistical analysis both for WRF and for CHIMERE has been done calculating the statistics for each station individually and the averaging all station together so that e.g., the 5 values of the individual relative humidity NRMSE are averaged to produce the final NRMSE value for the domain. The calculation has been done on the original hourly values from observations and model outputs and consider hourly values from the model only if the corresponding hourly observation is present. According to comments made by reviewer 4 and 5, MNB and RMSE have been substituted by MFB and MFE in the validation of WRF and CHIMERE.

*Q. It would be helpful to specify how wind direction statistics were calculated. Since wind direction is a circular variable, calculating means, RMSE, etc. is different than for linear variables. Also, I'm not sure that normalized measures, MNB, NRMSE make sense for wind direction.*

A > The statistics presented originally in the manuscript has been calculated as follows:

$$MNB = \frac{\sum_{i=1}^{n}(M_i - O_i)}{\sum_{i=1}^{n}(O_i)}$$

$$RMSE = \sqrt{\frac{\sum_{i=1}^{n}(M_i - O_i)^2}{n}}$$

As the review suggests these operators can be used for linear variables such as temperature and relative humidity but they haven't the same meaning for what concern circular variables like in the case of the wind direction. Moreover, they rely also on the number of observations point included in the denominator and the final value can be misleading. For this reason, the statistical analysis in the

new manuscript has been changed and the MNB and RMSE values substituted with mean fractional bias and error (MFB and MFE) originally used only for the validation of CHIMERE. Moreover, for WRF we also use the Index of Agreement calculated as follows:

$$\text{IOA} = 1 - \left[ \frac{\sum_{i=1}^{n}(O-M)^2}{\sum_{i=1}^{n}(|M-\bar{O}|+|O-\bar{O}|)^2} \right]$$

*Q. In the discussion of statistical evaluation of meteorological parameters it would be helpful to include criteria for what constitutes "good agreement" (line 361), "acceptable agreement" (line 443), etc.*

A > These qualitative terms have been deleted and the paragraphs modified to include quantitative statements.

*Q. Table 3 needs to include units for each meteorological variable. Shouldn't the MNB for KEN2K wind speed be negative?*

A > The statistics of Wind Speed for the domain KEN2K have been modified due to the presence in the observations of data from a station that after further analysis, was found to be suspect. In absence of precise information on the possible cause for this (the mean wind speed in that particular station was 45 m s$^{-1}$ with only few data available during the month) we have excluded that station from the statistical evaluation and performed the calculation again. The new values are in line with expectations.

*Q. There are conflicting statements about model performance for wind speed and direction. Lines 426-427 claim that the Nairobi and Kampala show higher agreement than Addis Ababa, but line 431 says that Kampala is the worst performing of the three cities.*

A > The conflicting statements have been addressed and the new paragraph can be found at lines **486-508.**

*Q. In Table 4, why are the Mean MOD PM$_{2.5}$ values different for Daily and Hourly? And it seems strange that the hourly NRMSE values are lower than the daily NRMSE.*

A > The Values in Table 4 have been checked and re-calculated. The initial difference between daily and hourly values of modelled PM$_{2.5}$ was present in Nairobi (roadside site) and Addis Ababa. The reason of the mismatch was related to the treatment of the Nan values by the python code used for the calculation of the statistics. The issue has been addressed and the averages re-calculated.

*Q. In Figure 8 the data for Nanyuki show what appears to be a nearly constant baseline PM$_{2.5}$ concentration of around 2 to 2.5 µg m$^{-3}$. Why would this be occurring?*

A > The observations used to validate CHIMERE performance for Kenya comes from previous work by Pope et al., 2018 [1]. In that work the site of Nanyuki was chosen as rural spot in a location of minimum local air pollution influence. The data from Nanyuki has been used for the calculation of the net urban increment subtracting the rural background concentrations of Nanyuki from the urban concentrations in Nairobi.

The average concentrations around 2 µg m$^{-3}$ in the period between the 4$^{th}$ and the 11$^{th}$ are the levels of the rural background in absence of any external influence from meteorological parameters and in absence of local sources. The peak of concentrations visible is the other days are between 4 and 15 µg/m$^3$ that is in any case a low value in comparison with the concentrations from the urban area.

The difference in the baseline concentrations is given by the big difference between the days with possible transport of pollutants from days where this phenomenon is not visible, but it is exaggerated by the low scale of the concentrations (0-16 µg m$^{-3}$)

*Q. The PM$_{2.5}$ data from Figure 9 and Figure 8 don't seem to agree. For the period March 3 – March 10, Figure 9 reports a daily concentration of 53-55 µg m$^{-3}$each day. In Figure 8, however, the hourly concentrations for that same time period hover around 2.2 µg m$^{-3}$and never exceed 4 µg m$^{-3}$.*

A > PM$_{2.5}$ data provided in Figure 8 are hourly concentrations comparing model outputs and concentrations. PM$_{2.5}$ provided in Figure 9 were daily cumulative totals obtained by summing all the concentrations observed in Nanyuki during each day. The average concentrations during the period between the 4$^{th}$ and the 11$^{th}$ of March are around 2.2 µg m$^{-3}$, 2.2 * 24 hours = 52.8 µg m$^{-3}$. The Figure has been modified and the cumulative concentrations substitute with daily average concentrations to not confuse the reader and provide more consistent information.

*Q. In presenting data table results, the text is often mainly just stating the values that are already shown in the tables. (e.g., sections 3.1.2, 3.2.1, 3.2.2) These sections could be condensed and/or modified to include additional description and discussion of what the data values mean. For example, why might model performance for wind speed and wind direction vary for airport vs. urban locations, why is there such a strong correlation between model and observation in Nanyuki, what are possible reasons for differences in model performance between the different domains?*

A > The reasons for the different behaviour in the model performance between Kenya, Uganda and Ethiopia has been explained in the text with a new paragraph between line **586-593** *"The performance of CHIMERE vary between the domains of Kenya, Uganda and Ethiopia. The performance of the model has been optimised during the validation for the simulation of hourly concentrations of PM$_{2.5}$ in Kenya and the same configuration applied to the domain of Uganda and Ethiopia to compare the reliability of the model. The difference in performance can be connected to different reasons: In first place, the difference in the sampling methods used for the two sites in Kenya against the measurements taken in the U.S. Embassies of Kampala and Addis Ababa. A second element of differentiation can be connected to the location of the observation sites in the cases of the U.S. Embassies and/or the possible influence of local sources not accounted in the emission inventories."*. The motivation for the higher agreement between model and observation in Nanyuki has been added with a new paragraph at lines **599-604**: *"Finally, the site of Nanyuki is the location where the agreement between model and observations is highest. This site was chosen by Pope et al. (2018) as rural spot in a location of minimum local air pollution useful to calculate the net urban increment subtracting the rural background concentrations of Nanyuki from the urban concentrations in Nairobi. Is therefore intended by their work that the average concentrations in that site were low. The model reproduces this low level or contamination close to the reality and to results to be able to reproduce also peaks of contamination in particular days of February probably generated elsewhere (see Section 3.2.2)"*. The difference in the location of the observations can have an influence on the change in directions and speed recorded in the observations. These rapid changes given by the open spaces (airport) or the building canyon effects of urban areas cannot be captured by WRF that take in account of a coarser differentiation of the land use between urban and not urban areas. New paragraph in line **501-508**.

***Technical corrections:*** *Throughout the manuscript the authors mention "low air quality index". This could be interpreted as a low numerical value of the air quality index, indicating good air quality, but from the context it seems the authors are instead describing poor, or low, air quality. It would be better to use a different word than "low".*

*I found the initial paragraphs of section 2, Materials and Methods, to be unnecessary (lines 104-127). They provide a partial summary of emissions, observational data, and model simulations, but since it is not a complete description, the reader is left with many questions. There are separate subsections that do provide all the pertinent details and they are much easier to follow. I would suggest removing lines 104-127 and incorporating that information into the subsections as appropriate.*

A > following the suggestion of the reviewer the initial paragraph of section 2 Material and Methods has been shortened and the content already mentioned elsewhere deleted.

*Q. Clarify in section 2.2 that CHIMERE is run only for the 6x6 and 2x2 km domains. This is somewhat implied in the discussion of boundary conditions (line 174), but not clearly specified. It is not until section 3 (lines303-304) that it becomes clear.*

A > The clarification that CHIMERE has been used to run only the 6 km and 2 km domains has been added at the beginning of the paragraph that now says: *"The configuration adopted in this work to run simulations over the domain at 6 and 2 km uses initial and boundary conditions from the global three-dimensional chemistry-transport model (LMDz-INCA, (Hauglustaine et al., 2004)), both for gaseous pollutants and for aerosols for the most external domain at 6 km of resolution while for the most internal domains at 2 km of resolution, the boundary conditions are calculated from model outputs of the parent domains."* Lines **157-164**.

*Q. It might be clearer to mention the rural Nanyuki, Kenya site (lines 285-287) immediately following the description of the urban Nairobi, Kenya site (lines 280-281).*

A > The paragraph has been modified in agreement to the suggestion: *"For the city of Nairobi, data from urban background were provided by the roadside site at Tom Mboya Street in Nairobi (1.28° S, 36.82° E) while data from rural background were provided by the site of Nanyuki, Kenya (0.01° N, 37.07° E) both obtained from the field sampling campaign performed by Pope et al., (2018)."* Lines **300-303**.

*Q. Symbols and text on figures with maps (Fig 1, Fig 3, Fig 7, Fig 11) were too small to read without zooming in to at least 200%. I was left searching for tiny triangles, numbers, and coloured dots.*

A > Figure 1 has been modified in order to make the markers on three cities more visible to reader. Figure 3 has been modified and the size of each map increased in order to make all the observation points more visible to the reader. Figure 7 has been deleted because it provided the same information visible in Figure 3 and finally Figure 11 has been modified to make the coloured indications and labels more evident for the reader.

*Q. Table 2 – units on Elevation written as "(m a. g. l)" are unclear. I interpret this as meters above ground level. "Above ground level" would imply monitoring sites aloft. I suggest just using "(m)" because elevation of ground sites can be assumed as height above sea level.*

The Table 2 unit has been modified accordingly.

*Q. Line 392 – does "small mean bias" refer to MB or NMB, and is it only for temperature peaks or all temperature measurements? There is similar ambiguity about the use of "mean bias" of relative humidity in line 406*

A > originally the sentence was referred to the absolute bias calculated between the observed mean values of model and observations. The statistical analysis has been modified and the MNB deleted from the analysis so that in the new manuscript when the word bias is used it's referred always to the absolute difference between mean observational and modelled values.

Line 414 – change "…sampled, a better …" to "…sampled. A better …"

A > The text has been modified accordingly.

Line 436 – not clear what is meant by "both observation sites". The text seems to be describing results at Addis Ababa which only has a single site.

A > The text has been modified accordingly.

*Q. Would be helpful to add a sentence after line 332, stating the frequency of weather station data observations. From Figure 4 it seems to range from every 3 hours to every 6 hours. Table 4 needs to include units for PM$_{2.5}$ (presumably micrograms/m3, but not specified).*

A > A new line has been added to specify the variable frequency of the observations: ". *While for the case of WRF observations with frequency variable from 3 to 6 hours are available from the UK Met Office database for different locations […]* "Lines **270-271** In Table 4 has been the units for PM$_{2.5}$ have been also added (µg/m$^3$)

*Line 524 - 526, Eq. (2) and Eq. (3) are missing parenthesis in the denominator around (Co+Cm)*

The parenthesis has been added in the new manuscript.

*Q. Table 5 is never referred to in the text.*

A > The table has been referred in the text in the paragraph of analysis of the general statistics of MFB and MFE.

*Q. Line 598-599. Figure 7 is mentioned twice, but text seems to be describing Figure 8. Figure 7 seems to be almost the same as Figure 3c. Not sure why it is needed.*

A > The paragraph refers to Figure 8, the text has been modified accordingly. Figure 7 has been deleted; the information was included already in figure 3c. A new figure has been introduced according to the suggestion of reviewer 4 to show additional hotspots of PM$_{2.5}$ concentrations inside the KEN2K domain.

*Q. Figure 9 should use a date format consistent with other figures in the paper. (Figure 9 writes dates as YYYY-MM-DD, while other figures use DD/MM/YY)*

A > Figure 9 has been modified accordingly.

*Q. Lines 674-680. Could also consider the impact of precipitation on particulate levels.*

A > The paragraph has been modified to take in account of the possible presence of precipitation modelled or observed in those days when we see and model big changes in concentrations in Nanyuki. The new paragraph is at line **737-747**.

*Q. Line 689, change "constituencies where analyzed" to "constituencies were analyzed"*

A > The text has been modified accordingly.

*Q. Line 690-691, text mentions relative population density and references Figure 11, but Figure 11 does not include any population density data.*

A > The text has been modified and correct.

*Q. May want to mention that the AQI levels are for hourly measurements, while the WHO limit used is for a daily average.*

A > The information has been added at line **773-776**.

*Q. In Figure 11, the concentration scale for the plot on the right is set at 50 μg m$^{-3}$, which is lower than the maximum concentrations. It would be better to use a scale that encompasses the entire concentration range.*

A > Figure 11 have been modified and the colour scale now range from 0 to 80 that is the highest mean modelled value found in the map.

*Q. Line 737-738, Figure caption mentions "top right" and "bottom right" maps, but only one is shown.*

A > The Figure caption has been modified accordingly.

**References:**

1.      Pope, F.D.; Gatari, M.; Ng'ang'a, D.; Poynter, A.; Blake, R. Airborne particulate matter monitoring in Kenya using calibrated low-cost sensors. *Atmospheric Chemistry and Physics* **2018**, *18*, 15403-15418, doi:10.5194/acp-18-15403-2018.

---

## Author Comment (AC2)

The paper presents a promising strategy for air quality forecast, which they have applied to large regions of three east-African countries including their capitals Kampala, Addis Ababa and Nairobi (with a special focus on Nairobi, capital of Kenya). Studies on air quality modelling are extremely scarce, so this work is, by itself novel, particularly in the fact that the authors present results for three large cities along with the corresponding data. The numerical methods used by the authors and their tools are exposed clearly and their results in terms of comparing model to observations are good (and sometimes extremely good).

However, I feel like there is more potential with this study if the authors would spend less time and space in presenting too many statistics (sometimes not relevant as I will discuss below for specific points). Statistical discussion should be refocused in model average, observation average, MFE and MFB which as the authors themselves recognize are better suited to their study than the metrics they use first. It is interesting that the model performs correctly relative to observations, but once this is done the reader feels he has the right not to see more statistics but to get insight on the physics and chemistry that are at play : if the model provides a relatively correct assessment of the PM time series, then it would be very interesting to know what the model tells us on the composition of PM (and therefore, possible, on its sources). Is it made of mineral dust, primary anthropogenic contaminants, SOA? If the model is correct on specific station, then we would like to see a map of contamination that it produces for the entire simulation domain (that of Fig. 7 for example): is Nairobi the maximum for the entire domain, are there other hot spots? How id pollution channelled – or not – between the mountain slopes? In its current shape, this article sometimes looks like a technical report on the feasibility of a particular forecast system for specific regions, which is not really what is expected from a research article. I think that with the additions above, this article could give many more indications on the specificities on Particulate matter composition in this region, and yield more interesting questions for future research. I feel this article will deserve publication because they obtain a great performance in reproducing pollution in areas where this has rarely been attempted; Once major changes are brought (making the statistical discussion more straightforward and give more scientific material from the model outputs), I feel that this may become a breakthrough article for air quality modelling in Africa.

A > The authors thank the reviewer for their interest in the manuscript and for the detailed review they made of its content and form. What the authors have tried to do in this work is to assess models for the simulations of meteorology and atmospheric chemistry over three urban domains within East Africa, for the first time at high resolution and using only available data from open-source provider and limited amount of field measurements. Hence, this paper wants to open the way for further and future scientific works focused on refining different aspect of the model configuration adopted here and to improve in this way the predictions made by the models. From this point of view the first novelty of this work is represented by the detailed analysis of the model performances simply because it is the first time that WRF and CHIMERE have been used to simulate PM$_{2.5}$ in this area of the world and at this resolution.

Forthcoming works will use this same configuration and will explore additional aspects of the atmospheric composition of East African conurbation focusing on the primary and secondary composition, transport effects and also fraction of elemental and black carbon in the PM. This type of analysis though requires an additional preparation and/or level of detail of the input data used for the simulations that are not in the aim of the present work. The presented work provides the necessarily starting point for these further studies by our and other groups. The authors thank the reviewer to suggest the possibility to investigate the formation of secondary aerosols, it is a hot topic in the analysis of the PM$_{2.5}$ levels within urban conurbations. The information we have at the moment for

the creation of primary emissions of PM$_{2.5}$ are though limited to the lumped species and few additional components of it (Organic and Elemental carbons) that would require many assumptions in the reliability of the secondary components modelled by CHIMERE. What we have done instead according to the reviewer suggestion is to substitute the original Figure 7 with a new Figure (9) providing the average concentrations of PM$_{2.5}$ in the whole domain KEN2K and giving the position of additional hotspots of concentrations outside Nairobi urban area.

Finally, our future efforts are oriented in refining the configuration used for the models and also increase the level of detail of the input used for the simulations. This means of course to obtain more information about local sources of anthropogenic emissions but also quantify with higher detail the biogenic emissions that have surely an impact on the levels of air pollutants in urban and rural locations.

**MAJOR COMMENTS**

A. Statistics on wind

l. 362: it is unclear what is meant by wind direction » and its unit. The vocabulary used i not appropriate for wind direction (« higher » wind direction has no meaning in my opinion). Mean Normalized bias has no clear meaning either in this sense (if as I think wind direction is in degree). Authors should explain how they build their indicators for wind. For calculating the RMSE and MNB, how do they account for the difference between a wind oriented at 1° and one at 359°? they are close but the difference between them is large. In general, MNB and RMSE are not adapted to deal with angles. Even the average does not make sense (average between 1° and 359° is 180° which does not make any sense…). I suggest that the authors remove the statistics they have done for wind direction (and possibly replace them by a more meaningful way to do these statistics, e.g., average and standard deviation of the geometric angle between observed and modelled wind speed). An alternative is to rely mostly on Fig. 5.

A > The wind direction shown in the manuscript is defined in degrees and the text has been modified according to a more appropriate description of the variation of this variable. The statistical parameters of MNB and RMSE are calculated as shown in the following equations:

$$MNB = \frac{\sum_{i=1}^{n}(M_i - O_i)}{\sum_{i=1}^{n}(O_i)}$$

$$RMSE = \sqrt{\frac{\sum_{i=1}^{n}(M_i - O_i)^2}{n}}$$

We agree with the reviewer that this type of metric can be suitable for linear variables such as temperature and relative humidity, but they cannot be appropriate for the analysis of circular variables such as wind directions. The statistical evaluation of WRF has been therefore re-written using mean fractional bias and errors (MFB and MFE) for the weather variables and also using the calculation of the index of agreement (IOA):

$$IOA = 1 - \left[\frac{\sum_{i=1}^{n}(O-M)^2}{\sum_{i=1}^{n}(|M-\bar{o}|+|O-\bar{o}|)^2}\right]$$

The new statistics has been added to table 3 and a new discussion of these parameters for the 4 variables has been added in the manuscript.

B. Error on wind speed?

Wind speed seems largely and critically false for the Kenya domain (Table 3). Could the authors double-check their numbers?

A > The statistics of Wind Speed for the domain KEN2K have been modified due to the presence in the observations of data from a particular station that after further analysis, was found to be suspect. In absence of precise information on the possible cause for this (the mean wind speed in that particular station was 45 m s$^{-1}$ with only few data available during the month) we have excluded that station from the statistical evaluation and performed the calculation again. The new statistics have been expressed in terms of MFB, MFE, R and IOA.

C. Use the same metrics throughout the paper

l. 528: here the authors argue (insightfully in my opinion) that MFB and MFE have less problems than MNB and NRMSE which they use above. Why not use MFB and MFE throughout the paper? MFB and MFE could be calculated in time-average foe all stations in the first place, given in table 4, and Table 4 could be used to discuss the results relative to the Boylan and Russel criteria. This would be less confusing than the current presentation and would avoid needing Fig. 6 which is an unusual figure style and, lacking the time dimension, does not bring much more understanding to the reader.

A > Thanks for the valuable comment on this aspect. The reviewer is right is highlighting the possibility to use the same metrics of MFB and MFE both for the evaluation of WRF and for the evaluation of CHIMERE. These two metrics are definitely more appropriate than the classic use of MNB and RMSE that can be indicative for the linear variables but misleading for circular variables like wind direction.

According to the comments of the reviewer great part of the statistical analysis WRF has been re-written using as new metrics MFB, MFE and also index of agreement calculated as follows:

$$\text{IOA} = 1 - \left[ \frac{\sum_{i=1}^{n}(O-M)^2}{\sum_{i=1}^{n}(|M-\bar{O}|+|O-\bar{O}|)^2} \right]$$

For what concern Figure 6 and the representation of CHIMERE performance in terms of MFB and MFE we think that that type of figure showing the capability of the model to reproduce PM$_{2.5}$ concentrations both in low and in high peak of concentrations represent an important information at this stage of the numerical modelling of air quality in east Africa. One of the elements of uncertainties in this work is represented by the use of low spatial resolution anthropogenic emissions to simulate urban and rural concentrations compared with observation points. The reliability of the model needed to be evaluated in the representation of both the general hourly pattern of concentrations but also in the discrimination of period of low and high concentrations with the available data for simulations. In the vision of using WRF-CHIMERE as tool to assist policy making these are level of reliability needed and important to address.

D. What do the authors mean by « coupled models »?

Usually, coupled modelling means that the chemistry-transport model is able to give feedback to the meteorological model (both models running at the same time, similar to a general circulation

model with atmosphere + ocean). This is not the case here, so the expression « coupled models » is confusing and should be removed from the text.

A > The reviewer is right is pointing out the imprecise definition of coupled model. The authors reviewed this term along the manuscript and modify it accordingly. The WRF and CHIMERE models have been used independently one from the other. Output from the simulations of WRF have been used off-line as weather parameters for the CHIMERE simulations.

l. 75: lon-term → long-term

A > this typo (like all the other found in the manuscript) has been corrected.

l. 148: is it really two-way nesting (do the small domains retroact on the large one?)? Otherwise, this is one-way nesting.

A > Yes, it is. The two-way nesting is a proper option activable on in the WRF model that let the parent and nest domain communicate the variable values and give modification retroactively from the nested to the parent domain.

l. 343: why such a massive underestimation for temperature in Kenya? Such a difference would strongly affect photochemistry isn't it?

A > The value of 4.1 °C is a big difference in temperature but calculated on the average of all stations for Kenya. One station in particular, Narok shows a huge difference between model and observations (10.9 °C) and this big value influence the average bias on all the stations. The bias in temperature for the station of Nairobi is 1.3°C smaller. This difference between the value of the single station of Narok and its influence on the average of the stations has been added in the explanation of the statistics (Lines **391-400**).

l. 392 MNB is not really significant here, it would depend on if temperature is expressed in K or °C. In any case, 0.1 MNB is not small at all (relative to an average value of 20° this is a 2° bias which is not small). I see something strange in the numbers presented in Table 3. There is a negative bias of 4.1° relative to an observed mean of 23.2° so with the typical definition of the mean bias ( ( - ) / ) I would expect a MNB ~0.18 while the authors claim the MNB is 0.1 here. This looks underestimated. I am aware of the difference between NMB and MNB and I don't have the data to recalculate the MNB here (see the definitions in e.g., https://www.bnl.gov/envsci/schwartz/pres/metrics.pdf). Could the authors explain what exact definition they have retained for the mean normalized bias and how they deal with missing data? This all the more surprising as for UGA2K and ETH2K I find exactly the same number as in Table 3 by calculating - ) / . This is also an argument to just drop MNB which behaves in a confusing way, as suggested in my major comment C.

A > The values of 4.1°C is the MNB calculated on the average of all the 5 stations for Kenya where the station of Narok in particular has a bias of 10.9 °C between model and observations. The absolute bias calculated for the station of Nairobi as previously explained is 1.3°C and it is related to the MNB calculated in the same station that is -0.06. In the same way the other number 0.1 is the MNB calculated for the station of Addis Ababa. The paragraph has been modified with and this aspect clarified.

The values of MNB and RMSE calculated for the statistics take in account the absence of the observation points in the calculation. The model-observation values are taken in account in calculation only when the observation is present for that particular hour.

l. 453: « negligible » is not the correct appreciation for biases up to 4°C in temperature. Table 4 : The figures don't always make sense. For Addis Ababa, there is a MNB 0.1 for daily data but 0.88 for hourly data, this is much likely affected by data points with an extremely small denominator driving the entire average. If the model yields 2.0 and 2.0, and if the observations yield 0.1 and 3.9 then the MNB would be 0.5 * (1.9 / 0.1 – 1.9 / 3.9) ~10 which hardly makes any sense. Again, see my major comment C.

A > The paragraphs of the WRF validation have been modified considering the new statistical parameters inserted in the analysis. Albeit there is a bias in the averaged station statistics to take in account the individual stations, the closest stations to the urban areas of interest shows acceptable biases and levels of MFB and MFE inside the range of acceptability of the model.

Fig. 6 : This figure is not really useful, just the statistics on MFB and MFE would be sufficient for the understanding. I do not think their distribution as a function of concentration really brings something to the discussion

A > As already explained in point C of the reviewer analysis, Figure 6 and the representation of CHIMERE performance in terms of MFB and MFE give an insight of the capability of the model to reproduce $PM_{2.5}$ concentrations both in low and in high peak of concentrations. The reliability of a numerical model for Air quality purposes needs to be evaluated in the representation of both the general hourly pattern of concentrations but also in the discrimination of period of low and high concentrations with the available data for simulations.

Fig. 8 : please use the same x-axis for both panels to ease reading the figure

A > The x-axis of Figure 8 has been modified and it is now in the same format of the other Figures.

Fig. 9 : It is not useful to compare modelled values in Nanyuki to observed values in Nyeri, 60km away in a mountain / plateau environment. No statistical link between the two timeseries can be expected a priori. I do not understand the point of the authors here, this should maybe be explained more.

A > The analysis of concentrations observed in Nanyuki takes in account that the location chosen by Pope et al. (2018) for the sampling of PM was a rural spot in a location of minimum local air pollution chosen to calculate the net urban increment subtracting the rural background concentrations of Nanyuki from the urban concentrations in Nairobi. The comparison that is proposed by Figure 8 it is only one of the options that can be taken in account considering the combined effect of meteorological parameters and location with higher contamination levels near Nanyuki that could influence the local level of PM. A first element to take in account to explain the peaks of contamination in Nanyuki could be the presence of local sources not accounted in the emission inventory used in CHIMERE. Despite this there is a clear change of trend in the concentration levels between February and March, in presence of local sources misrepresented we should see peaks at high concentration also in March but instead they are absent. A second element to take in account is the possible presence of precipitation during the period of March were the average concentrations of $PM_{2.5}$ doesn't exceed the 2 µg/m$^3$ but (Pope et al., 2018) affirm In their work that no rain was observed in that period and WRF model also doesn't model any in that particular period.

We are aware that to support the thesis of transport phenomena additional further analysis (e.g., trajectory analysis) are required as well as more observational point along the way between Nyeri

and Nanyuki. Further analyses are planned to go in that direction, what we argue in this paper is to give a possible explanation with the extent of the data available at the moment.

l. 643 : the authors claim that Nyeri is 0.43°N but n 10 on Fig. 7 seems to be at 0.43°S (or at least clearly nor 0.43°N)

A > The typo in the manuscript has been modified according to the suggestion of the reviewer.

**References:**

1. Pope, F. D., Gatari, M., Ng'ang'a, D., Poynter, A., and Blake, R.: Airborne particulate matter monitoring in Kenya using calibrated low-cost sensors, Atmospheric Chemistry and Physics, 18, 15403-15418, 10.5194/acp-18-15403-2018, 2018.

---

## Author Comment (AC3)

*Q. There is no discussion on why the modelling was performed on 2 km x 2 km resolution. Given the uncertainty in the models, their input data, and the usage of the results this extra step of modelling may only have introduced extra uncertainty compared to, say, retaining the 11 km x 11 km of the emissions inventories used. The authors shall also explain why they only simulated a 1-month period and not a full seasonal cycle.*

A. The performance of the modelling system needs to be compared with observations from the real world to assess their reliability. These observations are taken in different environments that can have different characteristics (e.g., rural, urban, or roadside backgrounds) and be affected by potential local sources. The necessity to bring the model prediction at high resolution is connected to the representation of the concentrations inside each grid cell. In the case of simulations at coarse resolution e.g., the 6x6km domain, the value of concentration of the pollutant X is the average of an area of 36km$^2$ and difficult to be representative of observations taken in a particular place. On the other hand, increasing the spatial resolution and bringing it to 2x2 km the average value inside each grid cell will be representative of a smaller area such as 4 km$^2$. WRF and CHIMERE model work up to a maximum of spatial resolution of 1km that represent the limit for which the numerical prediction can be considered reliable but there is a not negligible difference in computational time between 2x2 and 1x1km of resolution. The decision to simulate at final resolution of 2x2 is then the combination of higher representation of the concentrations modelled with observations, high reliability of the prediction and reasonable computational times. A new paragraph has been added in the manuscript explaining this at lines **536-551**.

The authors understand that the change of spatial resolution of the emissions could introduce uncertainty in the final emissions, but the tool used for the spatial allocation – "emisurf2016" – has been proven to be mass conservative and therefore the total amount of the emission in the original cells (original resolution) is the same than in the reallocated ones (2x2 km of resolution).

Finally, the period of simulation has been chosen in agreement with the available observations from another working strand of the same project. Observations or PM$_{2.5}$ from the analysed region for long period of times (e.g., one year) are almost impossible to find without large gap in observations due to problems in the infrastructures or not reliability of the sampling methods. Moreover, the assessment of a configuration of a modelling system can give better performance in particular season than in other due to combination of meteorological and chemical parameters.

Being this, this first work assessing a modelling system for atmospheric chemistry simulation in the East African area the author decided to limit the validation and application to a period of time for where a) continuous observations were present and b) the performance of the model could be validated with more precision. A new paragraph to clarify this aspect has been added in the text at line **268-281.**

*Q. In the description of the CHIMERE model there needs to be a short description on how wet- and dry deposition of gases and particles are treated.*

A. A description of the dry- and wet- deposition schemes included in CHIMERE have been added at Line **171-175** of the new manuscript.

*Q.I did not find the Validation of the WRF simulation particularly impressive, and I find the statement in the conclusion to be misleading "WRF has proved capable of reproducing the main meteorological patterns for all domains considered." (Line 762). For example, the overall temperature bias for the 1-month simulation is 4.1K in Nairobi – which, I believe, is the same order of magnitude as the seasonal*

*variation of monthly mean temperatures in Nairobi. I also lack an evaluation of the WRF-model's capability of reproducing precipitation in the different modelling domains.*

A > Thanks for this valuable comment. We agree that the statement as it was written was misleading and it has changed accordingly. The new line says: *"WRF showed a variable capability in reproducing the main surface weather patterns according to the different conditions of the three domains"* (Line **846-847**).

Despite this, the authors want to highlight that the validation of the WRF configuration has been done on three different domains with large differences in terms of topography and local weather conditions. The performance of the model has been assessed from the high and dry altitudes observation point in Ethiopia to locations near the Lake Victoria in Uganda. These environments are meteorological different and the validation of an individual configuration for WRF able to describe them (albeit in a defined seasonal period) represents a valuable and useful result.

The difference in absolute temperature of 4.1°C is calculated on the average of all the stations of Kenya and it is not relative to the station of Nairobi. The statistics of the individual stations (not shown in the paper) show that the bias for Nairobi is only 1.3°C. The highest bias is found for the station of Narok where the bias in temperature is around 10°C. A new paragraph explaining this has been added at line **391-400.**

The value of mean normalised bias (MNB) has been calculated considering the individual hourly data and excluding from the computation those hours not available from observation in order to have a more precise evaluation of the distance between model and data from the real world. This way of calculation has given us the possibility to exclude some periods where the observations were unavailable due to blackout in the networks or different technical issues.

It is right that a complementary evaluation of the performance of WRF should be done also taking in account rainfall. The manuscript has been modified mentioning this as an additional possibility for the differences in concentrations and observations in Nankuki but according to the reference material from Pope*, et al.* [1] no rainfall was observed in the simulated period.

Forthcoming works will be focused on the extension of the simulations over annual period and will take include the validation of the WRF configuration accounting also for rainfall events from the long- and short rain seasons in the equatorial countries.

Q. There are numerous typos throughout the manuscript.

A. The Manuscript has been carefully checked in all its parts and all the typos modified and corrected.

Q. Several of the maps provided would, in my opinion, be more useful if they, for example, show land surface type, population density, or emissions rather than height above mean sea level. The site indicators in Figs. 3, 7 and 11 need to be clearer.

A. The maps 1, 3 and 11 have been modified according to the suggestions of the reviewer. All the indicators have been increased in size, the text near made more visible and the size of the images increased in general. Maps 7 has been deleted because the information provided in Figure 3c were the same than in Figure 7. The height above the sea level is an information useful to understand the possible transport paths for the domain of Kenya and have been kept in Figure 3 for all three domains but the MODIS land use classification has been introduced for the three domains. The land use classification shows the different surface types present in the three domains.

Q. There are no units specified in Tables 3 and 4.

A. The units for the variables are Temperature (ºC), relative humidity (%), wind direction (degrees), wind speed (m s$^{-1}$) and PM$_{2.5}$ µg/m$^3$ and have been added to Tables 3 and 4.

Q The specified diurnal cycle of RH is wrong (row 405)

A. The line has been modified accordingly the new line says: *"Despite this both the diurnal peaks and night lowest values seems to be not correctly reproduced by the model that tends to overestimate the formers and underestimate the latter with a mean bias between -0.1 and 0.004."* Line **466-467**

*Q. The denominator of Eqs. (2) and (3) are in error*

A. The parenthesis has been added in the denominators of Eq. 2 and 3 in the new manuscript.

*Q. Explain the strict lower baseline of ca. 2 µg/m3 in modelled and observed PM2.5 in Fig. 8*

A. The observations used to validate CHIMERE performance for Kenya comes from a previous work made by Pope et al., 2018 [1]. In that work the site of Nanyuki was chosen as rural spot in a location of minimum local air pollution influence. The date from Nanyuki, in fact, in that work have been used for the calculation of the net urban increment subtracting the rural background concentrations of Nanyuki from the urban concentrations in Nairobi.

The average concentrations around 2 µg/m$^3$ in the period between the 4$^{th}$ and the 11$^{th}$ are the levels of the rural background in absence of any external influence from meteorological parameters and in absence of local sources. The peak of concentrations visible is the other days are between 4 and 15 µg/m$^3$ that are in any case lower in comparison with the concentrations from the urban area.

The difference in the baseline concentrations is given by the big difference between the days with possible transport of pollutants from days where this phenomenon is not visible, but it is exaggerated by the low scale of the concentrations (0-16 µg/m$^3$)

**References:**

1.      Pope, F.D.; Gatari, M.; Ng'ang'a, D.; Poynter, A.; Blake, R. Airborne particulate matter monitoring in Kenya using calibrated low-cost sensors. *Atmospheric Chemistry and Physics* **2018**, *18*, 15403-15418, doi:10.5194/acp-18-15403-2018.

---

## Editor Decision (ED1)

**I. Editor comments related to the response to the previous referee comments**

Referee #3:
1) Q. Not entirely clear how statistical measures are averaged (lines 336-338). Are measures calculated for each site and then those values for each domain are averaged (e.g., the 5 Relative Humidity NRMSE values for the 5 KEN2K weather stations are averaged to produce the KEN2K NRMSE value?) Or are the observed and modelled data for all the sites within a domain used together to calculate the average measure?
A > The statistical analysis both for WRF and for CHIMERE has been done calculating the statistics for each station individually and the averaging all station together so that e.g., the 5 values of the individual relative humidity NRMSE are averaged to produce the final NRMSE value for the domain. The calculation has been done on the original hourly values from observations and model outputs and consider hourly values from the model only if the corresponding hourly observation is present. According to comments made by reviewer 4 and 5, MNB and RMSE have been substituted by MFB and MFE in the validation of WRF and CHIMERE.
**Editor:** Is this information included in the text? If not, please do so.

2) Q. It would be helpful to specify how wind direction statistics were calculated. Since wind direction is a circular variable, calculating means, RMSE, etc. is different than for linear variables. Also, I'm not sure that normalized measures, MNB, NRMSE make sense for wind direction.
A > The statistics presented originally in the manuscript has been calculated as follows:
$MNB = \sum (Mi - Oi) \, n \, i{=}1 \sum (Oi) \, n \, i{=}1$
$RMSE = \sqrt{} \sum (Mi - Oi) \, n \, 2 \, i{=}1 \, n$
As the review suggests these operators can be used for linear variables such as temperature and relative humidity but they haven't the same meaning for what concern circular variables like in the case of the wind direction. Moreover, they rely also on the number of observations point included in the denominator and the final value can be misleading. For this reason, the statistical analysis in the new manuscript has been changed and the MNB and RMSE values substituted with mean fractional bias and error (MFB and MFE) originally used only for the validation of CHIMERE. Moreover, for WRF we also use the Index of Agreement calculated as follows:
$IOA = 1 - [\sum (O{-}M) \, n \, 2 \, i{=}1 \sum n \, (|M{-}\bar{O}|{+}|O{-}\bar{O}|)2 \, i{=}1 \,]$
**Editor:** Please add these equations to the manuscript, equivalently to Eqs. 2 and 3

3) Q. In the discussion of statistical evaluation of meteorological parameters it would be helpful to include criteria for what constitutes "good agreement" (line 361), "acceptable agreement" (line 443), etc.
A > These qualitative terms have been deleted and the paragraphs modified to include quantitative statements.
**Editor:** In lines 522 and 525, you still use 'acceptable' , in l. 410 'reasonable' without quantifying it. Please either avoid such statements or define them properly.
In particular, you may want to consider whether the paragraph at the end of section 3.1.2 is completely needed. – What do you want to say here – specify 'acceptable' for what.

4) Q. In Figure 8 the data for Nanyuki show what appears to be a nearly constant baseline PM2.5 concentration of around 2 to 2.5 µg m-3 . Why would this be occurring?
A > The observations used to validate CHIMERE performance for Kenya comes from previous work by Pope et al., 2018 [1]. In that work the site of Nanyuki was chosen as rural spot in a location of minimum local air pollution influence. The data from Nanyuki has been used for the calculation of the net urban

increment subtracting the rural background concentrations of Nanyuki from the urban concentrations in Nairobi. The average concentrations around 2 µg m-3 in the period between the 4th and the 11th are the levels of the rural background in absence of any external influence from meteorological parameters and in absence of local sources. The peak of concentrations visible is the other days are between 4 and 15 µg/m3 that is in any case a low value in comparison with the concentrations from the urban area. The difference in the baseline concentrations is given by the big difference between the days with possible transport of pollutants from days where this phenomenon is not visible, but it is exaggerated by the low scale of the concentrations (0-16 µg m-3 )

**Editor:** Please add the relevant information on baseline PM to the manuscript.

5) Q. In presenting data table results, the text is often mainly just stating the values that are already shown in the tables. (e.g., sections 3.1.2, 3.2.1, 3.2.2) These sections could be condensed and/or modified to include additional description and discussion of what the data values mean.

**Editor:** In the revised manuscript, there are still instances of such descriptive text only listing values that are reported in the table without any discussion. Condense such texts (e.g. l. 418 -443) and add more interpretation as it has been done around l. 400.

6) Technical corrections: Throughout the manuscript the authors mention "low air quality index". This could be interpreted as a low numerical value of the air quality index, indicating good air quality, but from the context it seems the authors are instead describing poor, or low, air quality. It would be better to use a different word than "low".

**Editor:** This comment was not addressed in the previous response. It is a fair concern as 'low air quality index' may be interpreted either as 'low index for air quality' or 'index for low air quality'. Please address it and replace 'low air quality (index)' by a less ambiguous expression.

**Referee #4:**
1) Introductory comment:
 .... In its current shape, this article sometimes looks like a technical report on the feasibility of a particular forecast system for specific regions, which is not really what is expected from a research article. I think that with the additions above, this article could give many more indications on the specificities on Particulate matter composition in this region, and yield more interesting questions for future research. I feel this article will deserve publication because they obtain a great performance in reproducing pollution in areas where this has rarely been attempted; Once major changes are brought (making the statistical discussion more straightforward and give more scientific material from the model outputs), I feel that this may become a breakthrough article for air quality modelling in Africa.

**Editor:** I agree with the referee that your article raises many important questions regarding air quality in Africa. I respect your response that your study is focused on presenting the model performance for a few locations but may be a considered a starting point for future analyses and additional aspects.  Your paper would indeed benefit if you added a few sentences along those lines towards the end of the paper as an outlook on further research question that should be explored in forthcoming studies.  This could be part of the conclusion section.

Fig. 9 : It is not useful to compare modelled values in Nanyuki to observed values in Nyeri, 60km away in a mountain / plateau environment. No statistical link between the two timeseries can be expected a priori. I do not understand the point of the authors here, this should maybe be explained more.
A > The analysis of concentrations observed in Nanyuki takes in account that the location chosen

by Pope et al. (2018) for the sampling of PM was a rural spot in a location of minimum local air pollution chosen to calculate the net urban increment subtracting the rural background concentrations of Nanyuki from the urban concentrations in Nairobi. The comparison that is proposed by Figure 8 it is only one of the options that can be taken in account considering the combined effect of meteorological parameters and location with higher contamination levels near Nanyuki that could influence the local level of PM. A first element to take in account to explain the peaks of contamination in Nanyuki could be the presence of local sources not accounted in the emission inventory used in CHIMERE. Despite this there is a clear change of trend in the concentration levels between February and March, in presence of local sources misrepresented we should see peaks at high concentration also in March but instead they are absent. A second element to take in account is the possible presence of precipitation during the period of March were the average concentrations of PM2.5 doesn't exceed the 2 µg/m3 but (Pope et al., 2018) affirm In their work that no rain was observed in that period and WRF model also doesn't model any in that particular period.
We are aware that to support the thesis of transport phenomena additional further analysis (e.g., trajectory analysis) are required as well as more observational point along the way between Nyeri and Nanyuki. Further analyses are planned to go in that direction, what we argue in this paper is to give a possible explanation with the extent of the data available at the moment.

**Editor:** I am not convinced that your response fully addresses the referee's concern. Please explain in the manuscript why the comparison as performed in Figure 8 is justified.

**II. Additional Editor comments:**

**A. Comments regarding content and structure**

l. 15: Add the model resolution here.

l. 219: (1) which conversion factor from organic carbon to aerosol mass was applied? (2) replace 'for' by 'with' (multiplied with...)

l. 220: Why is it assumed that PM2.5 is only composed of carbon-containing components? How about other compounds, such as sulfate etc?

l. 320 – 325: This paragraph is neither a result nor a discussion of your results. Therefore, either connect it better to the results or remove it, as it seems out of place and redundant here.

l. 327 - 375: This text is still a description of the methodology and therefore should be a subsection of Section 2. Lines 327 – 349 could be included in a subsection 'Statistical parameters'; l. 350 – 375 describes 'Model resolution and simulations'.

l. 585: ' CHIMERE 'better reproduces' than what?

l. 682 - 693: The model-observation comparison in Fig 7a shows clearly that the model tends to overestimate the PM2.5 concentration. If the emissions in the model were correct, one would expect the opposite trend – as you correctly describe, i.e. lower predicted values as compared to observations since the latter represent point measurements whereas the former are gird-averaged values.
However, in Figure 7a, there seems to be a period where model/observation agreement is particularly poor (~ 28/02 – 05/03) that shows a very distinct trend, opposite to the expected one. What was

different during this period? If indeed this discrepancy is due to an incomplete/inappropriate emission inventory in the model, can the characteristics of the air mass give a hint on the missing/wrong emissions as a function of air mass type/history?

Table 7: Is this table necessary? It provides the same information as in l. 794 – 798 and in some of the following lines. I suggest removing it as it is neither a result nor part of their discussion.

l. 847: The simulation of 'weather patterns' were not the main goal of this study but simulation of trends of air pollution.

**B. Technical comments (language, journal standards etc)**

l. 18: replace 'tool' by 'model'

l. 37: define 'WWP' and add it as database to reference list as detailed on the journal website https://www.atmospheric-chemistry-and-physics.net/submission.html#manuscriptcomposition

l. 43: Can you give a reference to the data base? – Add to reference list.

l. 54: add 'UN Habitat, 2017' to reference list

l. 98/99: A verb seems to be missing in this sentence.

l. 123: (1) Figures should be numbered according to their reference in the text. Since here Fig. 3 is cited before Figure 2, please change them accordingly. (2) remove 'a, b, c' here and in the remainder if the manuscript – see my comment below regarding 'panel 3d'.

l. 247 – 249: This sentence is quite convoluted given its rather simple message. How about "The emissions used in this work might not reflect the true values due to missing emission sources and the mismatch of the simulated time period and the date of the emission inventories. "

l. 276 – 278: I do not understand this sentence.

Table 2: (1) Spell out Latitude, Longitude, Elevation. (2) Use consistent terminology for latitude. Here you use – whereas later in the text, you specify S, N.

Figure 3: Remove 'd)' from the last panel. It is a legend and therefore does not need a label. In the caption, replace 'in table d' by 'in the legend'.

Table 3: Do not use random abbreviations in the table and caption. Spell out all words (obs., rel., ... ) or define them in the caption (e.g. relative humidity (RH) which then can be used as RH in the table).

l. 321: (1) remove 'from the real world'. (2) replace 'systems' by 'simulations'

l. 345 & 347: See my comment above regarding referring to Figures in the correct order. For simplicity, I suggest removing the text in the parentheses here. You can refer to it later.

l. 603: A subject is missing in this sentence (that starts with 'Is therefore...')

l. 721 and 729: Correct the units (m s$^{-1}$)

l. 756: 'and' seems wrong here ('and large hotspots...') – should it read 'a'?

l. 786: why 'e.g.'? Is 25 $\mu$g m$^{-3}$ a limit set by the WHO for comparable areas?
Table 6: (1) Replace 'WHO exceeding limit' by 'Exceedances of WHO limit'; (2) The last two columns do not include essential information: 'Ratio' is unclear and not very meaningful; 'model false positive' is described in the text and therefore does not need to be repeated here.

Data availability: Please provide at a minimum the model input and output data in a public repository https://www.atmospheric-chemistry-and-physics.net/policies/data_policy.html

1127 – 1129: Provide complete information for these references.

---

## Author Response (AR2)

**I. Editor comments related to the response to the previous referee comments**

Referee #3:

1) Q. Not entirely clear how statistical measures are averaged (lines 336-338). Are measures calculated for each site and then those values for each domain are averaged (e.g., the 5 Relative Humidity NRMSE values for the 5 KEN2K weather stations are averaged to produce the KEN2K NRMSE value?) Or are the observed and modelled data for all the sites within a domain used together to calculate the average measure?

A > The statistical analysis both for WRF and for CHIMERE has been done calculating the statistics for each station individually and the averaging all station together so that e.g., the 5 values of the individual relative humidity NRMSE are averaged to produce the final NRMSE value for the domain. The calculation has been done on the original hourly values from observations and model outputs and consider hourly values from the model only if the corresponding hourly observation is present. According to comments made by reviewer 4 and 5, MNB and RMSE have been substituted by MFB and MFE in the validation of WRF and CHIMERE.

**Editor: Is this information included in the text? If not, please do so.**

A >> A new subsection of Methodology has been introduced following the editor's comments: 2.5 Statistical parameters. Inside it this information has been clearly stated: *"The statistical analysis both for WRF and for CHIMERE has been done calculating the statistics for each station individually and the averaging all station together. The calculation has been done on the original hourly values from observations and model outputs and consider hourly values from the model only if the corresponding hourly observation is present."* **(Line 321 – 324)**

2) Q. It would be helpful to specify how wind direction statistics were calculated. Since wind direction is a circular variable, calculating means, RMSE, etc. is different than for linear variables. Also, I'm not sure that normalized measures, MNB, NRMSE make sense for wind direction.

A > The statistics presented originally in the manuscript has been calculated as follows:

$MNB = \sum (Mi - Oi) \, n \, i=1 \sum (Oi) \, n \, i=1$

$RMSE = \sqrt{} \sum (Mi - Oi) \, n \, 2 \, i=1 \, n$

As the review suggests these operators can be used for linear variables such as temperature and relative humidity but they haven't the same meaning for what concern circular variables like in the case of the wind direction. Moreover, they rely also on the number of observations point included in the denominator and the final value can be misleading. For this reason, the statistical analysis in the new manuscript has been changed and the MNB and RMSE values substituted with mean fractional bias and error (MFB and MFE) originally used only for the validation of CHIMERE. Moreover, for WRF we also use the Index of Agreement calculated as follows:

$IOA = 1 - [\sum (O-M) \, n \, 2 \, i=1 \sum n \, (|M-\bar{O}|+|O-\bar{O}|)2 \, i=1 \, ]$

**Editor: Please add these equations to the manuscript, equivalently to Eqs. 2 and 3**

A >> The equations relative to Pearson Coefficient R and Index of Agreement IOA have been added to the pre-existing equations of Mean Fractional Bias (MFB) and Error (MFE) in the new subsection 2.5 Statistical Parameters. The equations of MNB and RMSE haven't been added because these operations haven't been used in the final statistical analysis

3) Q. In the discussion of statistical evaluation of meteorological parameters, it would be helpful to include criteria for what constitutes "good agreement" (line 361), "acceptable agreement" (line 443), etc.

A > These qualitative terms have been deleted and the paragraphs modified to include quantitative statements.

**Editor: In lines 522 and 525, you still use 'acceptable' , in l. 410 'reasonable' without quantifying it. Please either avoid such statements or define them properly. In particular, you may want to consider whether the paragraph at the end of section 3.1.2 is completely needed. – What do you want to say here – specify 'acceptable' for what.**

A >> The sentence in l.410 has been modified as follows: *"The observed and modelled wind speeds in UGA2K, KEN2K and ETH2K suggest an overestimation by the model of 0.9, 0.8 and 0.2 m s$^{-1}$, respectively (Table 3)."*

The last paragraph at the end of section 3.1.2 has been modified as follows: *"Nevertheless, the more detailed analysis of the urban weather stations revealed discrepancies in the reproduction of relative humidity and wind direction for the station of Kampala (UGA2K) that could affect the deposition, removal and transport processes simulated by CHIMERE and will be object of future investigation to further improve the meteorological performance of WRF. Even if the bias found for some variable in the calculation of the averaged statistics over all stations was high, the individual weather stations close to the urban areas of interest showed smaller bias and levels of MFB and MFE inside the goal or criteria range of performance and therefore considered acceptable for simulations."* **(Line 526 – 532)**

4) Q. In Figure 8 the data for Nanyuki show what appears to be a nearly constant baseline PM2.5 concentration of around 2 to 2.5 μg m-3 . Why would this be occurring?

A > The observations used to validate CHIMERE performance for Kenya comes from previous work by Pope et al., 2018 [1]. In that work the site of Nanyuki was chosen as rural spot in a location of minimum local air pollution influence. The data from Nanyuki has been used for the calculation of the net urban increment subtracting the rural background concentrations of Nanyuki from the urban concentrations in Nairobi. The average concentrations around 2 μg m-3 in the period between the 4th and the 11th are the levels of the rural background in absence of any external influence from meteorological parameters and in absence of local sources. The peak of concentrations visible is the other days are between 4 and 15 μg/m3 that is in any case a low value in comparison with the concentrations from the urban area. The difference in the baseline concentrations is given by the big difference between the days with possible transport of pollutants from days where this phenomenon is not visible, but it is exaggerated by the low scale of the concentrations (0-16 μg m-3 )

**Editor: Please add the relevant information on baseline PM to the manuscript.**

A >> A new paragraph has been added at **line 709 – 714**: *"The site of Nanyuki was chosen by Pope et al. (2018) as rural spot in a location of minimum local air pollution influence. Data from Nanyuki was used for the calculation of the net urban increment subtracting the rural background concentrations of Nanyuki from the urban concentrations in Nairobi. On the one hand, the average concentrations around 3-4 μg m-3 in the period between the 4th and the 11th are, therefore, levels of the rural background in absence of any external influence from meteorological parameters and in absence of local sources."*

5) Q. In presenting data table results, the text is often mainly just stating the values that are already shown in the tables. (e.g., sections 3.1.2, 3.2.1, 3.2.2) These sections could be condensed and/or modified to include additional description and discussion of what the data values mean.

**Editor: In the revised manuscript, there are still instances of such descriptive text only listing values that are reported in the table without any discussion. Condense such texts (e.g. l. 418 -443) and add more interpretation as it has been done around l. 400.**

A >> The discussion relative to the statistical parameters has been modified and the made more descriptive of the numbers reported in the text. The paragraphs are at **line 406 – 451** have been modified accordingly.

6) Technical corrections: Throughout the manuscript the authors mention "low air quality index". This could be interpreted as a low numerical value of the air quality index, indicating good air quality, but from the context it seems the authors are instead describing poor, or low, air quality. It would be better to use a different word than "low".

**Editor: This comment was not addressed in the previous response. It is a fair concern as 'low air quality index' may be interpreted either as 'low index for air quality' or 'index for low air quality'. Please address it and replace 'low air quality (index)' by a less ambiguous expression.**

A >> the expression low air quality index has been modified with poor air quality index throughout all the manuscript.

Referee #4:

1)    Introductory comment:

.... In its current shape, this article sometimes looks like a technical report on the feasibility of a particular forecast system for specific regions, which is not really what is expected from a research article. I think that with the additions above, this article could give many more indications on the specificities on Particulate matter composition in this region, and yield more interesting questions for future research. I feel this article will deserve publication because they obtain a great performance in reproducing pollution in areas where this has rarely been attempted; Once major changes are brought (making the statistical discussion more straightforward and give more scientific material from the model outputs), I feel that this may become a breakthrough article for air quality modelling in Africa.

**Editor: I agree with the referee that your article raises many important questions regarding air quality in Africa. I respect your response that your study is focused on presenting the model performance for a few locations but may be a considered a starting point for future analyses and additional aspects. Your paper would indeed benefit if you added a few sentences along those lines towards the end of the paper as an outlook on further research question that should be explored in forthcoming studies. This could be part of the conclusion section.**

A >> A new paragraph has been added in the conclusion: *"The present work represents a first step in the use of numerical models for atmospheric chemistry simulations in East Africa with particular focus on urban conurbation. The aim of the present work was to assess the possibility to perform simulations with results close to observations in order to open the road for more detailed works. The natural next step of the present research aims to refine the quantity and quality of the input data used for the validation of both modelling system in order to improve the reliability of the predictions. Moreover, a more detailed analysis of the secondary inorganic and organic components of PM2.5 will be conducted for the three domains. Finally, the performance of CHIMERE will be tested in the reproduction of*

*gaseous species too in order to give a wider vision of the capabilities and opportunities of numerical modelling in this area of the world with present data."* **(Line 901 – 909).**

Fig. 9 : It is not useful to compare modelled values in Nanyuki to observed values in Nyeri, 60km away in a mountain / plateau environment. No statistical link between the two timeseries can be expected a priori. I do not understand the point of the authors here, this should maybe be explained more.

A > The analysis of concentrations observed in Nanyuki takes in account that the location chosen by Pope et al. (2018) for the sampling of PM was a rural spot in a location of minimum local air pollution chosen to calculate the net urban increment subtracting the rural background concentrations of Nanyuki from the urban concentrations in Nairobi. The comparison that is proposed by Figure 8 it is only one of the options that can be taken in account considering the combined effect of meteorological parameters and location with higher contamination levels near Nanyuki that could influence the local level of PM. A first element to take in account to explain the peaks of contamination in Nanyuki could be the presence of local sources not accounted in the emission inventory used in CHIMERE. Despite this there is a clear change of trend in the concentration levels between February and March, in presence of local sources misrepresented we should see peaks at high concentration also in March but instead they are absent. A second element to take in account is the possible presence of precipitation during the period of March were the average concentrations of PM2.5 doesn't exceed the 2 µg/m3 but (Pope et al., 2018) affirm In their work that no rain was observed in that period and WRF model also doesn't model any in that particular period. We are aware that to support the thesis of transport phenomena additional further analysis (e.g., trajectory analysis) are required as well as more observational point along the way between Nyeri and Nanyuki. Further analyses are planned to go in that direction, what we argue in this paper is to give a possible explanation with the extent of the data available at the moment.

**Editor: I am not convinced that your response fully addresses the referee's concern. Please explain in the manuscript why the comparison as performed in Figure 8 is justified.**

A >> A new paragraph has been added to make the motivations for the analysis of Figure 8 more robust and justifiable: *"The site of Nanyuki was chosen by Pope et al. (2018) as rural spot in a location of minimum local air pollution influence. Data from Nanyuki was used for the calculation of the net urban increment subtracting the rural background concentrations of Nanyuki from the urban concentrations in Nairobi. The average concentrations around 3-4 µg m-3 in the period between the 4th and the 11th are, on one hand, levels of the rural background in absence of any external influence from meteorological parameters and in absence of local sources.*

*On the other hand, the presence of higher hourly peaks in before and after the 4th to 11th can be linked to different reasons: the presence of local emission sources contributing to the peaks or the dispersion of polluted air masses from elsewhere towards the site of Nanyuki. It is important to observe that model and observations seems to agree particularly well in the description of the difference in magnitude between the different time periods excluding the possibility that the observed values can be influenced by local emission sources not accounted in the emission inventory. It seems more likely that those concentration levels are transported to Nanyuki from neighbouring areas with higher levels of PM2.5 contamination. To investigate this possible role of PM2.5 dispersion towards Nanyuki, we consider the closest MIDAS weather station to the sampling area of Nanyuki, in the town of Nyeri (0.43°S, 36.95°E altitude 1916 m a.g.l.) (n10 in Figure 2). Nyeri is only 60 km from the Nanyuki site and is situated between Mount Kenya (0.10°S, 37.30°E, altitude 4341 m a.g.l.) to the west and the Aberdare Range (0.46°S, 36.69°E, altitude 3441 m a.g.l.)."* **(Line 709 – 725)**

**II. Additional Editor comments:**

A. Comments regarding content and structure

**l. 15: Add the model resolution here.**

A >> Model resolution has been added.

**l. 219: (1) which conversion factor from organic carbon to aerosol mass was applied? (2) replace 'for' by 'with' (multiplied with...)**

A >> The conversion factor's value has been added and the text modified according to the suggestion.

**l. 220: Why is it assumed that PM2.5 is only composed of carbon-containing components? How about other compounds, such as sulphate etc?**

A >> The reason for the creation of PM2.5 using the carbon-containing component is motivated by the nature of the DICE emission Inventory that focuses on emissions particular diffuse and inefficient combustion emission sources (e.g., road transport, residential biofuel use, energy production and charcoal production and use). The variable PM2.5 created from DICE in this way has been then merged with the variable PM2.5 from EDGAR global emission inventory that contain information about inorganic fractions such as sulphate.

**l. 320 – 325: This paragraph is neither a result nor a discussion of your results. Therefore, either connect it better to the results or remove it, as it seems out of place and redundant here.**

A >> The paragraph has been deleted.

**l. 327 - 375: This text is still a description of the methodology and therefore should be a subsection of Section 2. Lines 327 – 349 could be included in a subsection 'Statistical parameters'; l. 350 – 375 describes 'Model resolution and simulations.**

A >> The manuscript has been modified accordingly and the information in line 327-375 moved in the new sections 2.5 Statistical Parameters and 2.6 Model resolution and simulations design

**l. 585: 'CHIMERE 'better reproduces' than what?**

A >> The text has been modified: *"Despite this and considering the daily average concentrations in the urban sites, the R coefficients were found to be between 30 and 42 % suggesting that CHIMERE better reproduces the concentrations of PM2.5 using daily than hourly values."* **(Line 590-592)**

**l. 682 - 693: The model-observation comparison in Fig 7a shows clearly that the model tends to overestimate the PM2.5 concentration. If the emissions in the model were correct, one would expect the opposite trend – as you correctly describe, i.e., lower predicted values as compared to observations since the latter represent point measurements whereas the former are gird-averaged values. However, in Figure 7a, there seems to be a period where model/observation agreement is particularly poor (~ 28/02 – 05/03) that shows a very distinct trend, opposite to the expected one. What was different during this period? If indeed this discrepancy is due to an**

**incomplete/inappropriate emission inventory in the model, can the characteristics of the air mass give a hint on the missing/wrong emissions as a function of air mass type/history?**

A >> Thanks for highlight this aspect in the simulation. The period between the 28[th] of February and the 5[th] of March do actually show hourly peaks of PM2.5 at high magnitude not found in the observation. A Similar event is present also in the period 10[th] 11[th] of March. The reason for this particular behaviour is not straightforward. Assuming that the observations from the field sampling are correct, one reason can be connected to the values of wind directions modelled by WRF towards that particular grid cell corresponding to Tom Mboya Street where in particular hours of the day additional amount of PM2.5 is moved from the from neighbouring cells. An additional explanation can be connected to the inorganic and organic secondary formation of PM2.5 that can contribute to the high hourly peaks modelled but not observed.

Nevertheless, the magnitude of the emissions in itself should not represent a problem because the average description of the minimum and maximum levels of PM2.5 modelled by CHIMERE show reasonable agreement with the observations both in urban area (where the emissions are supposed to be high) and in rural area (where the emissions are supposed to be low).

Despite this, as the reviewer suggests the magnitude of the emissions can be generally appropriate to the average levels of PM2.5 but can still be misrepresented at hourly level in urban area. It will be absolutely important to continue the work on refining the anthropogenic emissions used for this type of simulations and a possible step ahead is the creation of local emission inventory at high resolution able to increase the level of detail of the real levels of PM2.5 in urban areas.

**Table 7: Is this table necessary? It provides the same information as in l. 794 – 798 and in some of the following lines. I suggest removing it as it is neither a result nor part of their discussion.**

A >> Table 7 has been deleted.

**l. 847: The simulation of 'weather patterns' were not the main goal of this study but simulation of trends of air pollution.**

A >> The sentence has been modified substituting weather patterns with variables that have actually analysed for the validation of the model.

**B. Technical comments (language, journal standards etc)**

**l. 18: replace 'tool' by 'model'**

A >> The text has been modified accordingly.

**l. 37: define 'WWP' and add it as database to reference list as detailed on the journal website**

**https://www.atmospheric-chemistry-and-physics.net/submission.html#manuscriptcomposition**

A >> The reference has been added.

**l. 43: Can you give a reference to the data base? – Add to reference list.**

A >> The reference has been added.

**l. 54: add 'UN Habitat, 2017' to reference list**

A >> The reference has been added.

**l. 98/99: A verb seems to be missing in this sentence.**

A >> The text has been modified accordingly.

**l. 123: (1) Figures should be numbered according to their reference in the text. Since here Fig. 3 is cited before Figure 2, please change them accordingly. (2) remove 'a, b, c' here and in the remainder if the manuscript – see my comment below regarding 'panel 3d'.**

A >> The figures have been moved in the manuscript and the legend modified.

**l. 247 – 249: This sentence is quite convoluted given its rather simple message. How about "The emissions used in this work might not reflect the true values due to missing emission sources and the mismatch of the simulated time period and the date of the emission inventories. "**

A >> The text has been modified accordingly.

**l. 276 – 278: I do not understand this sentence.**

The sentence has been modified. The new sentence is: *"The period chosen for the simulations of meteorology has to be representative of the average weather conditions of the analysed area and avoid unusual weather conditions (e.g., extreme events) that could impact the physical and chemical processes described in the CTM and affect the final concentrations of secondary pollutants simulated"* **(Line 283-286).**

**Table 2: (1) Spell out Latitude, Longitude, Elevation. (2) Use consistent terminology for latitude. Here you use – whereas later in the text, you specify S, N.**

A >> The table has been modified accordingly.

**Figure 3: Remove 'd)' from the last panel. It is a legend and therefore does not need a label. In the caption, replace 'in table d' by 'in the legend'.**

A >> The image has been modified accordingly.

**Table 3: Do not use random abbreviations in the table and caption. Spell out all words (obs., rel., ...) or define them in the caption (e.g., relative humidity (RH) which then can be used as RH in the table).**

A >> The text has been modified accordingly.

**l. 321: (1) remove 'from the real world'. (2) replace 'systems' by 'simulations'**

A >> The text has been modified accordingly.

**l. 345 & 347: See my comment above regarding referring to Figures in the correct order. For simplicity, I suggest removing the text in the parentheses here. You can refer to it later.**

A >> The text has been modified accordingly.

**l. 603: A subject is missing in this sentence (that starts with 'Is therefore...')**

A >> The text has been modified accordingly.

**l. 721 and 729: Correct the units (m s-1)**

A >> The units have been modified accordingly.

**l. 756: 'and' seems wrong here ('and large hotspots...') – should it read 'a'?**

A >> The text has been modified accordingly.

**l. 786: why 'e.g.,'? Is 25 μg m-3 a limit set by the WHO for comparable areas?**

A >> The text has been modified accordingly.

**Table 6: (1) Replace 'WHO exceeding limit' by 'Exceedances of WHO limit'; (2) The last two columns do not include essential information: 'Ratio' is unclear and not very meaningful; 'model false positive' is described in the text and therefore does not need to be repeated here.**

A >> the text has been modified and the last two column of Table 6 deleted.

Data availability: Please provide at a minimum the model input and output data in a public repository https://www.atmospheric-chemistry-and-physics.net/policies/data_policy.html

A >> The emissions used for the simulations made by CHIMERE are available already in a public repository: https://doi.org/10.25500/edata.bham.00000695

The information relative to the availability of the other input data/models have been added in the section "Data Availability". This includes links for the download of both models and the observations of Meteorology. Observations of $PM_{2.5}$ for Nairobi are available upon request to Pope Francis and his team, while for Addis Ababa and Kampala are available upon request to the respective U.S. Embassies.

**1127 – 1129: Provide complete information for these references.**

A >> The references have been modified.